# ZeroQuant: Efficient and Affordable Post-Training Quantization for Large-Scale Transformers

**Zhewei Yao**\*, **Reza Yazdani Aminabadi, Minjia Zhang, Xiaoxia Wu, Conglong Li, Yuxiong He**

Microsoft
{zheweiyao, yazdani.reza, minjiaz, xiaoxiawu, conglong.li, yuxhe}@microsoft.com

## Abstract

How to efficiently serve ever-larger trained natural language models in practice has become exceptionally challenging even for powerful cloud servers due to their prohibitive memory/computation requirements. In this work, we present an efficient and affordable post-training quantization approach to compress large Transformer-based models, termed as ZeroQuant. ZeroQuant is an end-to-end quantization and inference pipeline with three main components: (1) a fine-grained hardware-friendly quantization scheme for both weight and activations; (2) a novel affordable layer-by-layer knowledge distillation algorithm (LKD) even without the access to the original training data; (3) a highly-optimized quantization system backend support to remove the quantization/dequantization overhead. As such, we are able to show that: (1) ZeroQuant can reduce the precision for weights and activations to INT8 in a cost-free way for both BERT and GPT-3-style models with minimal accuracy impact, which leads to up to 5.19x/4.16x speedup on those models compared to FP16 inference; (2) ZeroQuant plus LKD affordably quantize the weights in the fully-connected module to INT4 along with INT8 weights in the attention module and INT8 activations, resulting in 3x memory footprint reduction compared to the FP16 model; (3) ZeroQuant can be directly applied to two of the largest open-sourced language models, including GPT-J$_{6B}$ and GPT-NeoX$_{20B}$, for which our INT8 model achieves similar accuracy as the FP16 model but achieves up to 5.2x better efficiency.

## 1 Introduction

Large-scale natural language models have been widely adopted in different applications, e.g., natural language understanding using BERT [64] and generation tasks using GPT-style models [49]. Although those models have achieved cutting-edge accuracy results, as the model size keeps increasing dramatically, the requirements of memory footprint and the computational cost to deploy them become a major bottleneck, even on cloud servers with powerful GPU devices.

One promising way to alleviate this challenge is quantization, which can reduce the bit precision for both weight and activations for lower memory footprint and faster compute (e.g., INT8 Tensor cores on T4/A100). However, quantization usually requires retraining (also known as quantization aware training, or QAT in short) to recover the accuracy degradation from representation loss of weight and activations. To enable QAT, the full training pipeline is usually required, including the training data and compute resources, to finetune the model. Access to those components is now oftentimes not available, and QAT is also a time-consuming process, particularly for those large-scale models.

Recently, zero-shot quantization [10, 47] and post-training quantization (PTQ) [46, 39] are proposed to address the training-data access and compute requirement challenges since PTQ generally requires

---

*Code is released as a part of https://github.com/microsoft/DeepSpeed

36th Conference on Neural Information Processing Systems (NeurIPS 2022).

no (or minimal) retraining. But most of those works primarily focus on computer vision problems on relatively small scales. More recently, [7] shows promising PTQ results on BERT. However, (1) its main focus is on high-precision quantization (INT8/FP16) on $BERT_{base}$, (2) it does not consider other billion-scale generative models (GPT-3-style models [9]). More importantly, most of these works do not report real latency improvement, putting the usefulness of these methods in improving inference latency into question. For example, existing work often do not discuss the quantization/dequantization cost associated with different quantization schemes, which in fact has a big impact to the performance benefit of using low precision.

Besides, for extreme quantization (e.g., INT4), knowledge distillation is usually used to boost performance, which adds another source of expensive computation cost as compared to QAT. Furthermore, in order to achieve better accuracy performance, hidden-states knowledge distillation, e.g., [3, 81], is usually applied for the quantized model. This would put significant pressure on the GPU memory and the compute resource requirement since both the teacher and student models needed to be loaded into the GPU memory for training.

In this paper, we present ZeroQuant, an end-to-end post-training quantization and inference pipeline, to address those challenges, targeting both INT8 and INT4/INT8 mixed-precision quantization. Specifically, our contributions are:

- We apply fine-grained hardware-friendly quantization schemes on both weight and activations, i.e., group-wise quantization for weight and token-wise quantization for activations. Both quantization schemes can significantly reduce the quantization error and retain hardware acceleration properties.
- We propose a novel layer-by-layer knowledge distillation method (LKD) for INT4/INT8 mixed-precision quantization, where the neural network is quantized layer-by-layer through distillation with minimal iterations and even without the access to the original training data. As such, at any given moment, the device memory is primarily populated only with a single extra layer's footprint, making billion-scale model distillation feasible with limited training budget and GPU devices.
- We develop a highly optimized inference backend, which eliminates the expensive computation cost of quantization/dequantization operators, enabling latency speedups on INT8 Tensor cores on modern GPU hardware.
- Our empirical results show that:
  - ZeroQuant enables quantizing BERT and GPT-3-style models into INT8 weight and activations to retain accuracy without incurring any retraining cost. Compared to FP16 inference, our INT8 model achieves up to 5.19x/4.16x speedup on $BERT_{base}$/GPT-$3_{350M}$ on A100 GPUs.
  - ZeroQuant plus LKD can do INT4/INT8 mixed-precision quantization for BERT and GPT-3-style models. This results in a 3x memory footprint reduction with marginal accuracy loss as compared to the FP16 model. Also, thanks to the lightweight of LKD, we can finish the quantization process in 33s (10 minutes) for $BERT_{base}$ ($BERT_{large}$). We also demonstrate that LKD can use other datasets to achieve similar performance to the original training data.
  - We demonstrate the scalability of ZeroQuant on two of the largest open-sourced language models, i.e, GPT-$J_{6B}$ and GPT-NeoX$_{20B}$, with INT8 quantization. ZeroQuant can achieve 3.67x speedup over the FP16 model for GPT-$J_{6B}$ and (2) reduce the GPU requirement for inference from 2 to 1 and latency from 65ms to 25ms for GPT-NeoX$_{20B}$ (i.e., 5.2x better system efficiency in total).

## 2   Related Work

Model compression has been explored from different aspects [26, 38, 40, 35, 44, 21, 25, 51, 19, 76, 41, 27, 56, 60, 29, 61, 69, 34, 15, 39, 32]. Among those, quantization is one of the most promising directions as it directly reduces the memory footprint and compute intensity. Here, we focus on quantization for NLP models and briefly discuss the related work.

The majority of quantization works can be categorized into quantization-aware training (QAT). [57, 78] are the first few works to quantize BERT models using integer numbers for both weight and activations. Particularly, [57] utilizes Hessian information to push the weight bit-precision to even INT2/INT4, and it also proposes group-wise quantization to quantize the weight matrix in a more fine-grained granularity compared to single matrix quantization. [22] introduces quantization noise to alleviate the variations of QAT. [81, 3] leverage very expensive knowledge distillation [27] and data augmentation [29] to ternarize/binarize weights. [30] combines knowledge distillation [29] and learned step size quantization [20] to quantize the weight to 2–8 bits. Recently, [62] also uses

knowledge distillation to compress GPT-2 models on task-specific problems to INT2. All those works quantize models using the original training datasets. More importantly they need retraining or finetuning the full model to recover the accuracy, and such compute cost on extra-large models, like [58, 12], can be hardly affordable for most research labs or practitioners.

One solution to overcome the compute cost challenge is post-training quantization (PTQ). However, PTQ often induces a significant drop in accuracy because the network can be sensitive to quantization errors. Along this line, one of the first works applied to Transformer-based [65] models is [77]. The authors introduce centroid-based quantization method, where outlier numbers use FP32 format and the rest numbers are quantized using non-uniform quantization. As such, it is hard to get the real inference latency benefit on general compute accelerators, e.g., CPU and GPU, because the parallel processing units in these hardware do not support efficient computation of mixed data types. More recently, [7] introduces high-precision activation quantization (FP16) for part of the model to overcome the high dynamic activation ranges. However, to the best of our knowledge, (1) How to apply PTQ on GPT-3-style models while achieving high accuracy has not been studied in any of previous work yet; (2) How to apply PTQ on billion (or even a dozen of billions) scale model is still under-explored; (3) Efficient inference system backend is still missing, especially for fine-grained quantization schemes, making it hard to achieve low latency on commodity hardware. ZeroQuant resolves all those limitations by considering the system backend into the algorithm design and we verify its capability on both BERT and large-scale GPT-3-style (up to 20 billion, i.e., GPT-NeoX$_{20B}$) models for various tasks.

## 3 Background and Challenge

### 3.1 Transformer Architecture

The transformer architecture usually has three components: an embedding layer, a stack of encoder/decoder layers, and a final classifier. In this paper, we focus on quantizing the encoder/decoder layers, i.e., the transformer block, because it is often the most memory and compute intensive components in the entire architecture. With a transformer block, there are two sub-layers, the multi-head self-attention (MHSA) and the feed-forward connection (FFC). We give a short review later and please refer to [65] for more details. At high level, transformer models can be broadly categorized to three branches: encoder-only models (BERT) [64], decoder-only models (GPT-3-style) [49], and encoder-decoder models (T5) [50]. In this paper, we focus on encoder-only and decoder-only models but our approach can be applied to encoder-decoder models as well.

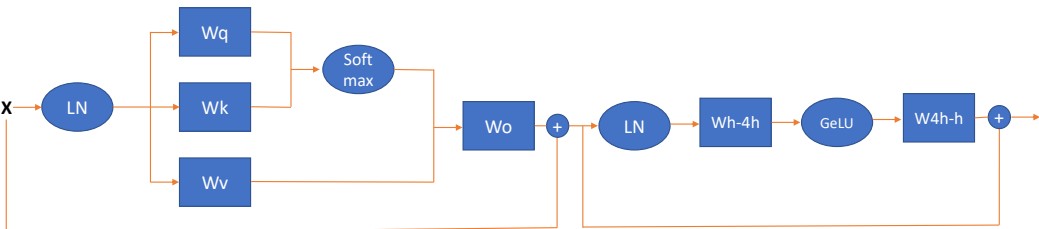

Figure 1: The illustration of a Transformer-block.

**Transformer Block** Assume the input of an encoder layer is $X$, the query, key, value, attention output, FFC dense, and FFC output matrices are $W_q$, $W_k$, $W_v$, $W_o$, $W_{h-4h}$, and $W_{4h-h}$, respectively. Then the forward propagation of a transformer-block is illustrated in Figure 1, where LN is the layer normalization, Softmax is the softmax operator, and GeLU is the activation function.

### 3.2 Quantization Background

Quantization maps high-precision numbers, e.g., FP16/FP32, to its low-precision counterpart, e.g., INT4/INT8, to reduce the model footprint and improve the compute performance. In this work, we use uniform symmetric scalar quantizers. That is to say, if we have a vector/matrix, $\mathbf{x}$, the quantization is applied as

$$\mathbf{x}_{quantize} = round\left(clamp(\frac{\mathbf{x}}{S}, -2^{bit-1}, 2^{bit-1} - 1)\right), \tag{1}$$

Table 1: Post training quantization results of GPT-3$_{350M}$ on 20 zero-shot evaluation datesets. Here WxAy means x-/y-bit for weight/activation. Particularly, for W4/8, we quantize the MHSA's weight to INT8 and FFC's weight to INT4. Please see Table I.1 for the results of all 20 tasks.

| Precision | Lambada (↑) | PIQA (↑) | OpenBookQA (↑) | RTE (↑) | ReCoRd (↑) | Ave. 19 Tasks (↑) | Wikitext-2 (↓) |
|---|---|---|---|---|---|---|---|
| W16A16 | 49.3 | 66.3 | 29.4 | 53.8 | 75.1 | 38.9 | 21.5 |
| W8A16 | 49.3 | 66.1 | 29.6 | 54.2 | 74.8 | 38.5 | 22.1 |
| W16A8 | 44.7 | 64.8 | 28.2 | 52.7 | 69.2 | 37.8 | 24.6 |
| W8A8 | 42.6 | 64.1 | 28.0 | 53.1 | 67.5 | 37.8 | 26.2 |
| W4/8A16 | 0.00 | 51.4 | 30.2 | 52.7 | 16.1 | 28.9 | 1.76e5 |

where $bit$ is the number of bit we use to represent the quantized value, and $S$ is the scaling factor. For weight matrix quantization, $S$ is generally computed as $S = max\,(abs(\mathbf{x}))$, since the weight matrix is static during inference. On the other hand, activations' range is dynamic during inference so that an accurate $S$ requires dynamic calculation during inference. However, to achieve best latency reduction, coarse-grained static quantization is usually applied in practice, where $S$ is calibrated using training data (e.g., momentum based averaging) and fixed during inference [24]. Although static quantization achieves better latency reduction, it also limits the quantization representation for activations, which is discussed below.

### 3.3 Post Training Quantization

Post-training quantization (PTQ) exhibits great compression efficiency compared to quantization-aware training (QAT) since PTQ is usually applied to quantize the model without retraining. A common strategy of PTQ is to feed the training data to the network and calibrate the scaling factor, $S$, using the running mean. Please see Appendix A.1 for more details.

Some work has been done for BERT$_{base}$ models [7] with INT8 weight and mixed INT8/FP16 activation quantization. However, there is no investigation for (1) even lower bit-precision PTQ on BERT models and (2) large-scale GPT-3-style models. Here, we briefly discuss the challenge of the application of PTQ on both BERT (in Appendix C) and GPT-3-style models.

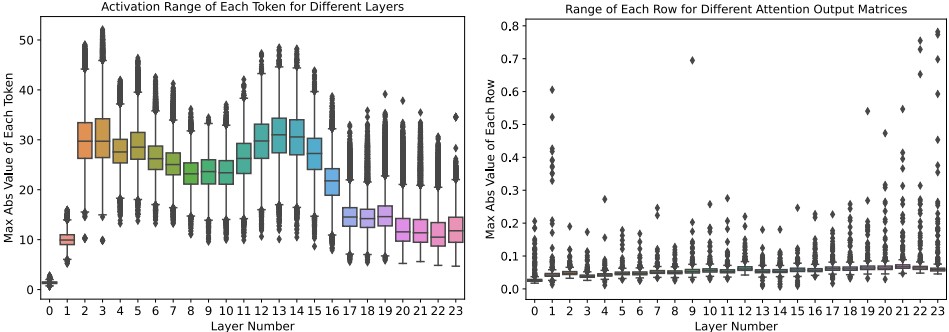

Figure 2: The activation range (left) and row-wise weight range of the attention output matrix (right) of different layers on the pretrained GPT-3$_{350M}$. See Figure C.1 for the results of BERT$_{base}$.

The results of GPT-3$_{350M}$ with PTQ are shown in Table 1. As can be seen, the INT8 activation quantization (i.e., the row of W16A8) causes the primary accuracy loss. Further pushing the weight to INT8 (i.e., the row of W8A8) does not change the accuracy of zero-shot evaluation tasks but leads the causal language modeling task (Wikitext-2) to worse perplexity score, which demonstrates the sensitivity of generation tasks as compared to other zero-shot evaluation problems. For W4/8A16, on some accuracy-based tasks, GPT-3$_{350M}$ still achieves reasonable performance like OpenBookQA but it loses accuracy on the majority of the rest tasks. Particularly, for Wikitext-2, GPT-3$_{350M}$ with W4/8A16 cannot generate any meaningful text anymore. Please also see Appendix C for the analysis for BERT.

**Dynamic Activation Range** To investigate why INT8 activation leads to significant accuracy drop for both BERT and GPT-3-style models, we plot the token-wise (i.e., the hidden state of each token) range of each activation for different transformer layers of GPT-3$_{350M}$ in Figure 2 (left). As can

be seen, different tokens have dramatically different activation ranges. For example, the maximum range of the last layer is around 35 but the minimum range is close to 8. This larger variance in the activation range makes it difficult to use a fixed quantization range (usually the maximum value) for all tokens to retain the prediction accuracy, because the limited representation power for small range tokens is going to hurt the accuracy performance.

**Different Ranges of Neurons in Weight Matrices** Similarly, we plot the row-wise (i.e., the output dimension) weight range of the attention output matrix ($W_o$) of GPT-3$_{350M}$ in Figure 2 (right). There is a 10x difference between the largest magnitudes of different rows and this leads to the worse generation performance of the INT8 weight PTQ. This also makes it very challenging when INT4 quantization is applied as the INT4 only has 16 numbers and a 10x smaller range leads to 2 (or 3) numbers for the representations of those smaller-range rows.

This analysis results also indicate why more expensive hidden-states knowledge distillation [3, 37] is used for ultra-low precision quantization to close the accuracy gap. However, as the training cost of knowledge distillation for large-scale models is too high, a lightweight and efficient method is desirable for PTQ.

# 4 Methodology

## 4.1 Fine-grained Hardware-friendly Quantization Scheme

As shown in Section 3, even applying INT8 PTQ to BERT/GPT-3-style models leads to significant accuracy degradation. The key challenge is the representation of INT8 cannot fully capture the different numerical ranges of different rows in weight matrices and different activation tokens. One way to address this is to use group-wise (token-wise) quantization for the weight matrix (activations).

**Group-wise Quantization for Weights** Group-wise weight matrix quantization has first been proposed in [57], where a weight matrix $W \in \mathbb{R}^{n \times m}$ is partitioned in to $g$ groups, and each group is quantized separately. However, in [57], the authors only apply this for quantization aware training. More importantly, they do not consider the hardware efficiency constraint and they do not have a system backend support. As such, they lack the real latency reduction benefit.

In our design, we consider the hardware constraint from Ampere Architecture of GPUs (e.g, A100), where the compute unit is based on Warp Matrix Multiply and Accumulate (WMMA) tiling size [54] to achieve the best speedup. Later, we will show that our group-wise quantization leads to much better accuracy as compared to single-matrix quantization due to its finer-granularity quantization while still achieving great latency reduction.

**Token-wise Quantization for Activations** As mentioned in Section 3 and Appendix 3.2, a common practice for existing PTQ work is to use static quantization for activation, where the min/max range is calculated at an offline calibration phase. Such a method might be sufficient for small scale models where the variance in the activation range is small. However, as analyzed in Section 3, there is a huge variance in the activation range for large-scale transformer models such as GPT-3$_{350M}$ and BERT$_{base}$. As such, a static quantization scheme (often applied to all tokens/samples) would lead to significant accuracy drop. One natural idea to overcome this issue is to adopt finer-grained token-wise quantization and dynamically calculate the min/max range for each token to reduce the quantization error from activations. Our evaluation in Section 5 also shows that token-wise quantization for activation significantly improves the accuracy of GPT-3-style and BERT models.

However, directly applying token-wise quantization using existing DL frameworks, such as the PyTorch quantization suite, would lead to significant quantization and dequantization cost because token-wise quantization introduces additional operations that lead to expensive data movement overhead between the GPU compute units and the main memory. To address this issue, we build a highly optimized inference backend for token-wise quantization of transformer models. For example, the inference backend of ZeroQuant employs so called *kernel fusion* technique to fuse quantization operator with its previous operator, like layer normalization, to alleviate the data movement cost from token-wise quantization. Similarly, the dequantization cost of the different GeMMs' output is alleviated by scaling the INT32 accumulation using both the weight and activation quantization scales, before writing the final FP16 result back to the main memory for the next FP16 operator (like GeLU). Those optimization will be discussed in more details in Section 4.3.

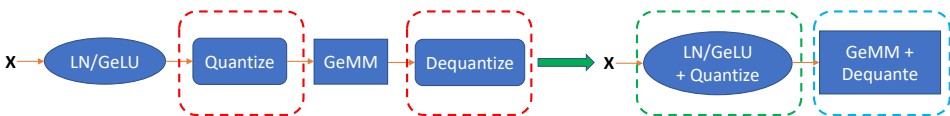

Figure 3: The illustration of normal (left) and our fused (right) INT8 GeMM.

Token-wise quantization can significantly reduce the representation error for quantized activations. Also, as it does not need to calibrate the activation range, later we will show that there is no quantization-related cost (e.g., activation range calibration) for a moderate quantization scheme (INT8 weight with INT8 activation) for ZeroQuant.

## 4.2 Layer-by-layer Knowledge Distillation with Affordable Cost

Knowledge distillation (KD) is one of the most powerful methods to alleviate the accuracy degradation after model compression. However, there are several limitations of KD, especially for hidden-states KD on large-scale language models: (1) KD needs to hold a teacher and a student model together during the training, which dramatically increases the memory and compute cost; (2) KD usually requires full training of the student model. Therefore, several copies (gradient, first/second order momentum) of the weight parameters need to be stored in memory to update the model; (3) KD generally requires original training data, which sometimes are not accessible due to privacy/confidential issues.

To address those limitations, we present our layer-by-layer distillation (LKD) algorithm. Assume the target model for quantization has $N$ transformer blocks, $L_1, ..., L_N$, the accessible dataset has input $(\boldsymbol{X}, \boldsymbol{Y})$, which can be the original training data or datasets from other resources. Our LKD quantizes the network layer-by-layer and uses its original (i.e., unquantized) version as the teacher model. More specifically, assume layer $L_k$ is going to be quantized, and its quantized version is $\widehat{L}_k$. Then we use the output of the $L_{k-1}$ (i.e., by running inference on $X$ over the first $k-1$ layers) as the input of $L_k$ and $\widehat{L}_k$, measure the difference, and do the model update to $L_k$, i.e.,

$$\mathcal{L}_{LKD,k} = MSE\left(L_k \cdot L_{k-1} \cdot L_{k-2} \cdot ... \cdot L_1(\boldsymbol{X}) - \widehat{L}_k \cdot L_{k-1} \cdot L_{k-2} \cdot ... \cdot L_1(\boldsymbol{X})\right), \quad (2)$$

where $MSE$ is the mean square loss, and it can be also replaced by other losses (e.g., KL divergence) as well. As can be seen, (1) our LKD does not need to hold a separate teacher as we use the same $L_1$ to $L_{k-1}$ for both teacher/student model. As such, the only extra model cost we have is $L_k$; (2) the memory overhead of optimizer states are significantly reduced as the only optimizing layer is $L_k$; (3) as we never optimize the end-to-end model, the training does not depend on the label anymore. Later, we will show that LKD does not rely on the original training data in Section 5.6.

## 4.3 Quantization-Optimized Transformer Kernels

Both optimizing the inference latency and model size is crucial for serving large-scale transformer models in practice. During inference, the batch size is often relatively small, so the inference latency of the model primarily depends on the time of loading inference needed data from the main memory. By quantizing the weights and activations to lower precision, we reduce the data volume needed to load those data, which allows more effective use of memory bandwidth and higher loading throughput. However, simply converting weights/activations to INT8 does not guarantee improved latency because there are additional data movement overhead associated with quantization/dequantization operations as shown in Figure 3 (red box). Such an overhead becomes expensive and in some cases surpasses the performance benefits of using low precision. To reap the accuracy improvement from token-wise quantization while obtaining improved latency, we now present our optimizations that maximize the memory bandwidth utilization to speed up inference latency for ZeroQuant.

**CUTLASS INT8 GeMM** To support INT8 computation, we use CUTLASS [6] INT8 GeMM implementation tuned for different batch sizes. Unlike standard GPU backend library, such as cuDNN, using CUTLASS allows us to more flexibly fuse quantization operation before and after GeMM to reduce kernel launching and data-movement overhead.

**Fusing Token-wise Activation Quantization** Token-wise quantization/dequantization introduce many additional operations that lead to extra data movement cost. To eliminate these cost, we use

Table 2: Result of BERT_base on the development set of GLUE benchmark (except WNLI). [57]+ uses 128 groups for weight matrix which is hard to get GPU acceleration. [7]* uses mixed INT8 and FP16 activation, and it directly reports the average metric of MNLI/MRPC/QQP/STS-B, which is basically the average of the two metrics we used for our runs.

| Precision (Method) | CoLA | MNLI-m | MNLI-mm | MRPC | QNLI | QQP | RTE | SST-2 | STS-B | Ave. | Ave. Time (s) |
|---|---|---|---|---|---|---|---|---|---|---|---|
| W16A16 (Baseline) | 59.72 | 84.94 | 85.06 | 86.27/90.57 | 92.15 | 91.51/88.56 | 72.20 | 93.23 | 90.06/89.59 | 83.95 | N/A |
| W8A8 [57] (QAT)+ | — | 83.91 | 83.83 | — | — | — | — | 92.83 | — | — | — |
| W8A8 [78] (QAT) | 58.48 | — | — | —/89.56 | 90.62 | —/87.96 | 68.78 | 92.24 | 89.04/— | — | — |
| W8A8 (QAT) | 61.21 | 84.80 | 84.64 | 83.82/88.85 | 91.29 | 91.29/88.28 | 71.12 | 92.89 | 88.39/88.18 | 83.37 | 2900 |
| W8A8 (PTQ) | 56.06 | 79.99 | 81.06 | 75.49/79.67 | 87.35 | 89.92/86.82 | 48.38 | 91.40 | 86.58/86.44 | 77.41 | 6 |
| W8A8/16 [7] (PTQ)* | 58.63 | 82.67 | 82.67 | 88.74 | 90.41 | 89.40 | 68.95 | 92.66 | 88.00 | 82.46 | Unknown |
| W8A8 (ZeroQuant) | 59.59 | 84.83 | 85.13 | 86.03/90.39 | 91.98 | 91.45/88.46 | 71.12 | 93.12 | 90.09/89.62 | 83.75 | 0 |
| W4/8A16 (PTQ) | 0.00 | 16.74 | 16.95 | 31.62/0.00 | 50.74 | 63.18/0.00 | 47.29 | 70.64 | 16.48/15.91 | 33.11 | 6 |
| W4/8A16 (ZeroQuant) | 57.29 | 82.69 | 83.27 | 84.56/88.40 | 90.04 | 86.52/79.49 | 70.76 | 92.78 | 88.46/88.61 | 81.65 | 0 |
| W4/8A16 (ZeroQuant-LKD) | 58.50 | 83.16 | 83.69 | 84.80/89.31 | 90.83 | 88.94/84.12 | 70.04 | 92.78 | 88.49/88.67 | 82.35 | 31 |
| W4/8A8 (ZeroQuant) | 56.69 | 82.46 | 83.06 | 84.07/88.03 | 90.13 | 87.04/80.50 | 70.76 | 92.78 | 88.07/88.44 | 81.55 | 0 |
| W4/8A8 (ZeroQuant-LKD) | 58.80 | 83.09 | 83.65 | 85.78/89.90 | 90.76 | 89.16/84.85 | 71.84 | 93.00 | 88.16/88.55 | 82.71 | 31 |

*kernel fusion* [68] to fuse quantization operation for activation with its previous element-wise and/or reduction operations such as bias-add, GeLU, and LayerNorm into a single operator, as illustrated by the green box in Figure 3. For the dequantization operation (e.g., dequantizing the integer output from the GeMM operator), we similarly fuse it with our custom GeMM schedule to avoid additional read/write accesses to the main memory as illustrated by the blue box in Figure 3.

By doing the above optimizations, we are able to show significant latency reduction for BERT and GPT-3-style models in Section 5. Please see Appendix D for more details about our system optimization.

# 5 Results

**Experimental Details** To evaluate the proposed ZeroQuant, we test it on both BERT and GPT-3 models. For BERT, we tested both BERT_base and BERT_large on GLUE benchmark; and for GPT-3-style models, we tested the GPT-3_350M (i.e., GPT-3-style model with 350M parameters) and GPT-3_1.3B (i.e., GPT-3-style model with 1.3B parameters) on 20 zero-shot evaluation tasks, including 19 accuracy-based tasks and 1 language modeling generation task. To illustrate the scalability of the proposed ZeroQuant, we also directly apply it to two of the largest open-sourced GPT-3-style models, i.e., GPT-J_6B [67] and GPT-NeoX_20B [5]. We use a fixed set of hyperparameters for all the LKD-related experiments even though tuning them may benefit our results. Please see Appendix A.2 for more training details and see Appendix A.3 for the reported metrics for BERT. To provide a comprehensive study, we also include a tuning result in Appendix E on BERT and an ablation study for different proposed components in Section 5.5.

**Notation Explanation** We use WxAy to represent using x-bit for weight quantization and y-bit for activation quantization. Unless specific explanation, for W4/8, we quantize the MHSA's weight to INT8 and FFC's weight to INT4; for A8/16, we use FP16 activation for self-attention calculation (i.e., the GeMM related to $W_{q/k/v}$) and use INT8 for the rest calculation. We use ZeroQuant to represent the method with only fine-grained quantization schemes and use ZeroQuant-LKD to represent the method with both fine-grained quantization schemes and LKD.

A summary of results is shown in Appendix B. We also include (1) the results required by reviewers in Appendix G; (2) the speedup explanation .

## 5.1 Main Results of BERT

**BERT_base** We report the results of BERT_base in Table 2. For W8A8, the average accuracy of PTQ degrades more than 10 points. However, ZeroQuant can achieve 83.75 scores, which is only 0.2 lower than baseline. Particularly, as ZeroQuant has no activation range calibration phase, the cost of ZeroQuant is 0 which is even cheaper than standard PTQ. As compared to [7], our method achieves a better average score (1.29 higher). Meanwhile, as compared to INT8 activation used in ZeroQuant, [7] uses mixed INT8 and FP16 activation.

We also compare our method with our internal trained QAT and other QAT works [57, 78]. As can be seen, with comparable accuracy results as those QAT methods, ZeroQuant can save the retraining cost from 2900s to 0s for INT8 quantization.

Table 3: Result of BERT$_{large}$ on the development set of GLUE benchmark (except WNLI). [+]We extensively tuned the learning rate for QAT (see Appendix F for more details).

| Precision (Method) | CoLA | MNLI-m | MNLI-mm | MRPC | QNLI | QQP | RTE | SST-2 | STS-B | Ave. | Ave. Time (s) |
|---|---|---|---|---|---|---|---|---|---|---|---|
| W16A16 (Baseline) | 63.35 | 86.65 | 85.91 | 87.99/91.62 | 92.24 | 91.08/88.08 | 74.01 | 93.46 | 90.34/90.11 | 85.03 | N/A |
| W8A8 [78] (QAT) | — | — | — | —/90.9 | 91.74 | | | | 90.12/— | — | — |
| W8A8 (QAT)[+] | 59.85 | 86.65 | 86.35 | 85.29/89.43 | 92.55 | 91.60/88.60 | 61.37 | 93.23 | 87.55/87.65 | 82.78 | 7181 |
| W8A8 (PTQ) | 60.57 | 75.69 | 76.94 | 81.13/84.93 | 88.49 | 84.04/74.35 | 46.93 | 91.74 | 62.75/55.77 | 73.54 | 31 |
| W8A8 (ZeroQuant) | 63.38 | 86.52 | 85.64 | 87.75/91.50 | 92.31 | 91.09/88.05 | 72.56 | 93.35 | 90.45/90.19 | 84.81 | 0 |
| W4/8A16 (PTQ) | 0.00 | 16.85 | 33.24 | 68.38/80.89 | 51.25 | 63.18/0.00 | 52.71 | 52.41 | -5.74/-8.51 | 35.73 | 31 |
| W4/8A16 (ZeroQuant) | 62.99 | 84.77 | 84.42 | 87.50/91.16 | 91.63 | 90.03/86.41 | 48.01 | 92.16 | 89.49/89.28 | 81.23 | 0 |
| W4/8A16 (ZeroQuant-LKD) | 63.72 | 84.90 | 84.81 | 87.99/91.39 | 91.45 | 90.34/86.92 | 51.62 | 92.43 | 89.46/89.29 | 81.85 | 550 |
| W4/8A8 (ZeroQuant) | 62.34 | 84.62 | 84.25 | 87.75/91.38 | 91.87 | 89.86/86.09 | 47.65 | 91.97 | 89.39/89.17 | 81.06 | 0 |
| W4/8A8 (ZeroQuant-LKD) | 63.51 | 84.70 | 84.71 | 88.73/91.99 | 91.73 | 90.25/86.74 | 49.82 | 92.09 | 89.34/89.08 | 81.62 | 550 |

Table 4: Post training quantization result of GPT-3$_{350M}$ on 20 zero-shot evaluation datasets. Please see Table I.1 for the results of all 20 tasks.

| Precision (Method) | Lambada (↑) | PIQA (↑) | OpenBookQA (↑) | RTE (↑) | ReCoRd (↑) | Ave. 19 Tasks (↑) | Wikitext-2 (↓) | Time Cost |
|---|---|---|---|---|---|---|---|---|
| W16A16 | 49.3 | 66.3 | 29.4 | 53.8 | 75.1 | 38.9 | 21.5 | N/A |
| W8A8 (PTQ) | 42.6 | 64.1 | 28.0 | 53.1 | 67.5 | 37.8 | 26.2 | 7 mins |
| W8A8 (ZeroQuant) | 51.0 | 66.5 | 29.2 | 53.4 | 74.9 | 38.7 | 21.7 | 0 |
| W4/8A16 (PTQ) | 0.00 | 51.4 | 30.2 | 52.7 | 16.1 | 28.9 | 1.76e5 | 7 mins |
| W4/8A16 (ZeroQuant) | 10.1 | 58.5 | 27.2 | 52.0 | 56.5 | 33.5 | 88.6 | 0 |
| W4/8A16 (ZeroQuant-LKD) | 39.8 | 63.8 | 29.4 | 53.1 | 70.1 | 37.0 | 30.6 | 1.1 hours |
| W4/8A8 (ZeroQuant) | 10.5 | 57.7 | 28.0 | 52.7 | 55.3 | 33.4 | 92.1 | 0 |
| W4/8A8 (ZeroQuant-LKD) | 37.4 | 61.8 | 28.2 | 53.1 | 68.5 | 36.6 | 31.1 | 1.1 hours |

For the more aggressive weight quantization with minimal (or no) training quantization, i.e., W4/8A16, PTQ fully loses all accuracy (pure random prediction). However, ZeroQuant can still achieve an 81.65 average score. On top of ZeroQuant, if we add our LKD, the accuracy can be further boosted to 82.35 with a cost of 31s per task using only a single GPU, which is 93.5x cheaper than INT8 QAT quantization. We also test ZeroQuant and ZeroQuant-LKD under the W4/8A8 quantization scheme and both of them achieve similar accuracy performance as W4/8A16. If hyper-parameter tuning is applied to LKD, ZeroQuant-LKD can achieve an 83.22 average score under W4/8A8, which is similar to QAT's W8A8 result. Please see Appendix E for more details.

**BERT$_{large}$** We test our methods on BERT$_{large}$ as well and the results are shown in Table 3. Similar to BERT$_{base}$, ZeroQuant achieves much better accuracy than PTQ methods. As compared to QAT methods, ZeroQuant has comparable results on larger datasets (like MNLI/QQP) and has better performance on small tasks (e.e., CoLA/MRPC/RTE). We actually tune QAT for multiple learning rates but cannot get even better performance for those small tasks (see Appendix F for more details).

For more aggressive quantization schemes, like W4/8A16 and W4/8A8, ZeroQuant and ZeroQuant-LKD still achieve good accuracy except for RTE but the model size is about 3x smaller than FP16 counterpart. This is aligned with the INT8 QAT results, which lose significantly more accuracy on RTE. Thanks to the lightweight cost of LKD, it only takes about 550s to finish each task even on BERT$_{large}$, which is 13x cheaper than QAT.

## 5.2 Main Results of GPT-3-style Models

**GPT-3$_{350M}$** We first test ZeroQuant and ZeroQuant-LKD on GPT-3$_{350M}$ and report the result in Table 4. The first interesting finding of zero-shot evaluation on GPT-3-stype models is that the accuracy performance of accuracy-based tasks is more tolerant to quantization than generation tasks. For instance, W8A8 PTQ has a 1.1% average accuracy drop on 19 accuracy-based tasks as compared to 4.7 points loss on Wikitext-2. Comparing ZeroQuant with PTQ using W8A8, we can reduce the accuracy gap from 1.1% to 0.2% and the perplexity (PPL) gap from 4.7 to 0.2 with no activation range calibration cost.

For W4/8A16 quantization scheme, PTQ can hardly predict reasonable answers for the majority of tasks and its generation performance on Wikitext-2 is fully crashed. As a comparison, ZeroQuant still achieves non-trivial performance on some tasks but its generation performance significantly degrades on Wikitext-2. LKD brings a significant performance boost for this W4/8A16 setting. Note that ZeroQuant-LKD increases the accuracy from 33.5 to 37.0 and decreases the PPL from 88.6 to 30.6 compared to ZeroQuant, and the entire cost of this is just 3.1 hours on a single A100 GPU. Note that this is about 0.027% GPU hours of the full pretraining cost (128 A100 GPUs for 32 hours). Similar to W4/8A16, ZeroQuant-LKD achieves much better performance than ZeroQuant on W4/8A8 by using the lightweight LKD.

Table 5: Post training quantization result of GPT-3$_{1.3B}$ on 20 zero-shot evaluation datasets. Please see Table I.2 for the results of all 20 tasks.

| Precision (Method) | Lambada (↑) | PIQA (↑) | OpenBookQA (↑) | RTE (↑) | ReCoRd (↑) | Ave. 19 Tasks (↑) | Wikitext-2 (↓) | Time Cost |
|---|---|---|---|---|---|---|---|---|
| W16A16 | 61.3 | 71.4 | 33.6 | 53.1 | 82.6 | 42.4 | 15.3 | N/A |
| W8A8 (PTQ) | 54.8 | 67.7 | 16.6 | 54.5 | 75.7 | 40.5 | 18.9 | 13 mins |
| W8A8 (ZeroQuant) | 62.6 | 70.7 | 33.4 | 52.7 | 80.9 | 42.3 | 15.7 | 0 |
| W4/A16 (PTQ) | 0.00 | 50.4 | 27.0 | 50.9 | 15.8 | 29.0 | 1.35e5 | 13 mins |
| W4/8A16 (ZeroQuant) | 43.9 | 66.5 | 30.0 | 52.7 | 77.3 | 39.38 | 21.9 | 0 |
| W4/8A16 (ZeroQuant-LKD) | 59.4 | 69.5 | 31.6 | 52.7 | 79.7 | 41.5 | 17.6 | 3 hours |
| W4/8A8 (ZeroQuant) | 46.8 | 66.4 | 28.8 | 52.7 | 76.2 | 39.24 | 24.1 | 0 |
| W4/8A8 (ZeroQuant-LKD) | 48.7 | 68.1 | 29.0 | 52.0 | 77.4 | 39.90 | 18.2 | 3 hours |

Table 6: The speedup of our W8A8 as compared to W16A16. We measure the end-to-end average latency for the entire BERT model, and the time reported is in milliseconds.

| Seq Len BS | Precision | 128 | | | | | | | | 256 | | | | | | | |
|---|---|---|---|---|---|---|---|---|---|---|---|---|---|---|---|---|---|
| | | 1 | 2 | 4 | 8 | 16 | 16 | 64 | 128 | 1 | 2 | 4 | 8 | 16 | 16 | 64 | 128 |
| BERT$_{base}$ | W16A16 | 2.45 | 3.22 | 3.85 | 5.51 | 9.96 | 17.93 | 34.25 | 67.08 | 3.13 | 4.05 | 5.70 | 10.55 | 19.27 | 36.69 | 71.75 | 140.0 |
| | W8A8 | 1.08 | 1.16 | 1.42 | 1.76 | 2.58 | 3.90 | 6.74 | 12.92 | 1.22 | 1.44 | 2.08 | 2.88 | 4.10 | 7.80 | 14.66 | 28.13 |
| | Speedup | 2.27 | 2.78 | 2.71 | 3.13 | 3.86 | 4.60 | 5.08 | 5.19 | 2.57 | 2.81 | 2.74 | 3.66 | 4.70 | 4.70 | 4.89 | 4.98 |
| BERT$_{large}$ | W16A16 | 5.45 | 6.38 | 8.73 | 13.88 | 26.34 | 48.59 | 92.49 | 183.4 | 6.39 | 8.94 | 14.66 | 27.99 | 51.94 | 98.78 | 195.9 | 384.5 |
| | W8A8 | 2.08 | 2.58 | 2.84 | 3.79 | 6.21 | 10.28 | 18.86 | 36.62 | 2.55 | 3.36 | 4.16 | 6.88 | 11.61 | 21.20 | 41.24 | 79.90 |
| | Speedup | 2.62 | 2.47 | 3.07 | 3.66 | 4.24 | 4.73 | 4.90 | 5.01 | 2.51 | 2.66 | 3.52 | 4.07 | 4.47 | 4.66 | 4.75 | 4.81 |

**GPT-3$_{1.3B}$** The results of GPT-3$_{1.3B}$ are shown in Table 5. Similar to GPT-3$_{350M}$, for W8A8, Zero-Quant has much better performance than PTQ with less no activation calibration cost, particularly for the generation task Wikitext-2 (3.2 points lower). Also, for W4/8 quantization, LKD can bring non-trivial performance gain for ZeroQuant. The cost of LKD is about 0.02% of the full pre-training cost (128 A100 GPUs for 120 hours)

## 5.3 Latency Reduction of BERT and GPT-3-style Models

We compare the inference speed of BERT between FP16 and our INT8 versions in Table 6 on a single 40G-A100 GPU. Using our efficient quantization kernel implementation and operator fusion, the INT8 model can achieve 2.27–5.19x speedup on BERT$_{base}$ and 2.47–5.01x on BERT$_{large}$.

We also include the latency comparison of GPT-3-style models between FP16 and our INT8 version. Particularly, we use the model to generate the first 50 tokens based on a given text and measure the average latency. Our INT8 model leads to 4.16x/4.06x speedup for GPT-3$_{350M}$/GPT-3$_{1.3B}$ as compared to the FP16 counterpart.

## 5.4 A Showcase of GPT-J$_{6B}$ and GPT-NeoX$_{20B}$

To demonstrate the scalability of ZeroQuant, we applied it to two of the largest open-sourced models, i.e., GPT-J$_{6B}$ and GPT-NeoX$_{20B}$, which have 6B and 20B parameters separately.

We report the results of GPT-J$_{6B}$ in Table 7 on three generation datasets, i.e., PTB [42], Wikitext-2, and Wikitext-103 [43]. As can be seen, as compared to FP16 precision, ZeroQuant achieves similar PPL on all three different tasks. To compare the latency, we again use the average latency number to generate the first 50 tokens. Our W8A8 can get up to 3.67x speedup compared to the FP16 version.

To quantize GPT-NeoX$_{20B}$ to W8A8 for all GeMMs, the accuracy significantly decreases. We retrieve the quantization of each weight matrix and of each activation, and finally find out that the activation quantization for the attention calculation (i.e., the input of self-attention) causes the accuracy loss. We conjecture that this is because of the sensitivity of the self-attention module for extra-large models (20B) but cannot verify this for other models due to the lack of open-sourced extra-large models and the full evaluation pipeline. As such, we leave the input activation for self-attention in FP16 and quantize the rest to INT8. The results are shown in Table 8. Our W8A8/16 achieves similar accuracy performance but can reduce both the GPU resource requirement (from 2 A100 GPUs to 1) and the latency from 65ms to 25ms, which together lead to 5.2x better throughput/efficiency.

## 5.5 Ablation Study of Different Components

To investigate the performance gain of each component we introduced in Section 4, i.e., group-wise weight quantization, token-wise activation quantization, and lightweight layer-by-layer knowledge distillation, we here do an ablation study on BERT$_{large}$ with W4/8A8.

Table 7: Post training quantization result of GPT-$J_{6B}$ on three zero-shot generation tasks

| Precision | PTB | Wikitext-2 | Wikitext-103 | Latency |
|---|---|---|---|---|
| W16A16 | 20.47 | 10.35 | 10.35 | 29.13ms (1x) |
| W8A8 | 20.97 | 10.51 | 10.52 | 7.94ms (3.67x) |

Table 8: Post training quantization result of GPT-$NeoX_{20B}$ on 19 zero-shot evaluation datasets. Please see Table I.4 for the results of all 19 tasks.

| Precision | Lambada | PIQA | Ave. 19 Tasks | Latency |
|---|---|---|---|---|
| W16A16 | 71.7 | 77.7 | 50.5 | 2×65ms (1x) |
| W8A8/16 | 71.9 | 78.3 | 50.4 | 1×25ms (5.2x) |

Table 9: Ablation study of different components for $BERT_{large}$ on the development set of GLUE. The quantization scheme used here is W4/8A8. Here, GQ is the abbreviation of group-wise weight quantization, TQ is the abbreviation of token-wise activation quantization.

| GQ | TQ | LKD | CoLA | MNLI-m | MNLI-mm | MRPC | QNLI | QQP | RTE | SST-2 | STS-B | Ave. |
|---|---|---|---|---|---|---|---|---|---|---|---|---|
| ✗ | ✗ | ✗ | -0.79 | 33.07 | 32.94 | 68.38/80.54 | 49.42 | 63.18/0.00 | 52.71 | 52.29 | -4.27/-1.90 | 35.85 |
| ✓ | ✗ | ✗ | 59.81 | 66.63 | 68.79 | 68.63/71.17 | 83.87 | 78.24/61.30 | 46.93 | 89.45 | 54.58/32.52 | 66.52 |
| ✓ | ✓ | ✗ | 62.34 | 84.62 | 84.25 | 87.75/91.38 | 91.87 | 89.86/86.09 | 47.65 | 91.97 | 89.39/89.17 | 81.06 |
| ✓ | ✓ | ✓ | 63.51 | 84.70 | 84.71 | 88.73/91.99 | 91.73 | 90.25/86.74 | 49.82 | 92.09 | 89.34/89.08 | 81.62 |

Table 10: Post training quantization result of GPT-$3_{350M}$ on 20 zero-shot evaluation datesets The quantization scheme here is W4/8A8. Please see Table I.3 for the results of all 20 tasks.

| Method | Data Resource | Lambada (↑) | PIQA (↑) | OpenBookQA (↑) | RTE (↑) | ReCoRd (↑) | Ave. 19 Tasks (↑) | Wikitext-2 (↓) |
|---|---|---|---|---|---|---|---|---|
| ZeroQuant | — | 10.5 | 57.7 | 28.0 | 52.7 | 55.3 | 33.4 | 92.1 |
| ZeroQuant-LKD | Random data | 26.1 | 59.3 | 29.2 | 50.5 | 64.9 | 34.5 | 40.6 |
| ZeroQuant-LKD | Wikipedia | 33.9 | 62.4 | 28.0 | 52.7 | 69.5 | 36.2 | 30.4 |
| ZeroQuant-LKD | Original data | 37.4 | 61.8 | 28.2 | 53.1 | 68.5 | 36.6 | 31.1 |

We present the results in Table 9. As can be seen, group-wise weight quantization boosts the accuracy (random-guess prediction) from PTQ to a non-trivial result (66.52). Further adding token-wise quantization improves 14.54 points accuracy performance. On top of those (i.e., ZeroQuant), LKD further brings a 0.56 point gain.

## 5.6 No Access to The Original Training Data

As mentioned in previous sections, the original training data are oftentimes hard to access due to the privacy and/or confidential issues. Therefore, we here study the performance of our LKD when there is no direct access to the original training data. As the distillation objective of our LKD does not depend on the label, the training data used for LKD can be very flexible.

We compare the performance of GPT-$3_{350M}$ on W4/8A8 quantization scheme using three different training data resources, i.e., random data (using random integer number to generate token ids), Wikipedia (using Huggingface to get the data[2]), and original PILE dataset.

The results are shown in Table 10. Compared to ZeroQuant, LKD using random data can boost the accuracy by 1.1% and reduce the PPL from 92.1 to 40.6. The reason why random data can still significantly improve the performance is that LKD does not optimize the end-to-end pipeline and it only layer-by-layer learns the internal dependency from the teacher model. Therefore, random data can also provide meaningful information. Using Wikipedia data from Huggingface can further improve the accuracy to 36.2 and reduce the PPL to 30.4, which is comparable to the results using the original data. This indicates that a clean text dataset can be used for LKD when we do not have access to the original full dataset.

## 6 Conclusions

With the rapid growth of large model sizes, we have reach a point to consider how to serve those models in practice. Although several works demonstrate that post-training quantization can be applied to BERT models, to the best of our knowledge, there have been no existing works on (1) billion-scale GPT-3-style models, (2) ultra-low precision post-training quantization, and (3) end-to-end solution of how to efficiently serve the quantized model online. In this work, we offer fine-grained compression schemes for both weight and activations to enable INT8 quantization for up to 20B-scale models (GPT-$NeoX_{20B}$). We also offer a novel affordable layer-by-layer knowledge distillation for ultra-low precision quantization, which leads to 3x model size reduction compared to FP16 model while achieving minimal accuracy degradation. Furthermore, we provide a system backend support and show up to 5.19x speedup on BERT models and 5.2x better efficiency on GPT-$NeoX_{20B}$.

---

[2] https://huggingface.co/datasets/wikipedia

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
