# A  Experimental Details

## A.1  Details of PTQ on BERT and GPT

For BERT, we use a batch size of 32 and sequence length 128 to calibrate the range of activations. In order to capture the dynamic range, we use 0.95 momentum with 100 iterations, i.e.,

$$x_{max} = 0.95 x_{max} + 0.05 max(x_{current-iteration}),$$
$$x_{min} = 0.95 x_{min} + 0.05 min(x_{current-iteration}).$$

For GPT-3-style models, we use the same momentum method but change the batch size to 8 with sequence length 2048.

## A.2  Details of Main Result

**BERT**  BERT models are trained using the code-base from Huggingface [73]. We show our ZeroQuant method on $BERT_{base}$ and $BERT_{large}$. We use the same lower-case tokenizer in $BERT_{large}$ instead of the cased tokenizer in the original paper [16]. When fine-tuning on GLUE [66] tasks ((i.e., MRPC [18], STS-B [11], SST-2 [59], QNLI [52], QQP [28], MNLI [71], CoLA [70], RTE [14]).[3]), we follow the instruction from Huggingface Transformer Library [73].

For ZeroQuant and ZeroQuant-LKD, we use 48 groups for group-wise weight quantization on $BERT_{base}$ and 64 groups for group-wise weight quantization on $BERT_{large}$, for all the weight matrices.

For LKD, we use 100 iterations with batch size 32 and sequence length 128 for $BERT_{base}$, and we use 400 iterations for $BERT_{large}$. We fix the learning rate as 5e-6 for both models on all tasks. However, tuning them may favor ZeroQuant.

All the models are trained using a single 40G-A100 GPU (Azure ND A100 instances).

**GPT-3-style Models**  All GPT-3-style models used in the paper are trained using DeepSpeed [53] and Megatron-DeepSpeed Library [4]. The pretraining data are from PILE dataset [23], and the training pipeline and hyperparameters are based on the Megatron-DeepSpeed repository. We use 128 A100 GPUs (Azure ND A100 instances) to do the pretraining. It takes about 32 hours to finish the training of $GPT-3_{350M}$ and 120 hours of $GPT-3_{1.3B}$. We evaluate our results on 20 zero-shot evaluation tasks, including 19 accuracy evaluation tasks (i.e., HellaSwag [79], LAMBADA [48], TriviaQA [31], WebQS [4], Winogrande [55], PIQA [63], ARC (Challenge/Easy) [8], ANLI (R1/R2/R3) [72], OpenBookQA [45], RACE-h [33], BoolQ [13], Copa [1], RTE [14], WSC [36], MultiRC [75], ReCoRD [80]) and 1 language modeling generation task (i.e., Wikitext-2 [43]).

For ZeroQuant and ZeroQuant-LKD, we use 64/128 groups for group-wise weight quantization on $GPT-3_{350M}$/$GPT-3_{1.3B}$ for all the weight matrices.

For LKD, we use 1600 iterations with batch size 8 and sequence length 2048 for both $GPT-3_{350M}$ and $GPT-3_{1.3B}$. We fix the learning rate as 5e-6 for both models. However, tuning them may favor ZeroQuant.

All the quantized models are trained using a single 40G-A100 GPU (Azure ND A100 instances).

## A.3  Accuracy reported for BERT on GLUE

We report the performance metric for BERT on GLUE based on Table A.1. For the average score, if the task only has one metric, we use it for the final result; if the task has two metrics, we compute the average of the two metrics first and use it for the final average score. For instance, the score of MRPC used to compute the final average is the mean of its accuracy and F1 score.

# B  Summary of Main Results

The accuracy drop varies a lot under different benchmarks and scenarios. Here, we give a summary according to those settings:

---

[3]We exclude WNLI [36] since its results are not stable [17].
[4]https://github.com/microsoft/Megatron-DeepSpeed

Table A.1: Metric used for BERT$_{base}$ on the development set of GLUE benchmark (except WNLI).

| | CoLA | MNLI-m | MNLI-mm | MRPC | QNLI | QQP | RTE | SST-2 | STS-B |
|---|---|---|---|---|---|---|---|---|---|
| | Matthews Correction | Accuracy | Accuracy / F1 | Accuracy | Accuracy | Accuracy / F1 | Accuracy | Accuracy | Pearson / Spearmanr |

Table C.1: Post training quantization results of BERT$_{base}$ on development sets of the GLUE benchmark (except WNLI). Here WxAy means x-bit for weight quantization and y-bit for activation quantization. Particularly, for W4/8, we quantize the MHSA's weight to INT8 and FFC's weight to INT4. Please see Appendix A.3 for the reported metrics.

| Precision | CoLA | MNLI-m | MNLI-mm | MRPC | QNLI | QQP | RTE | SST-2 | STS-B | Ave. |
|---|---|---|---|---|---|---|---|---|---|---|
| W16A16 | 59.72 | 84.94 | 85.06 | 86.27/90.57 | 92.15 | 91.51/88.56 | 72.20 | 93.23 | 90.06/89.59 | 83.95 |
| W8A16 | 60.77 | 84.65 | 84.92 | 85.29/89.86 | 91.84 | 91.52/88.56 | 71.84 | 93.46 | 89.89/89.50 | 83.87 |
| W16A8 | 56.85 | 80.55 | 81.48 | 84.07/89.33 | 91.34 | 91.30/88.07 | 68.59 | 93.46 | 88.74/88.74 | 81.93 |
| W8A8 | 58.74 | 79.99 | 81.06 | 84.31/89.51 | 91.18 | 91.24/88.03 | 70.76 | 92.66 | 88.33/88.73 | 82.16 |
| W4/8A16 | 0.00 | 16.74 | 16.95 | 31.62/0.00 | 50.74 | 63.18/0.00 | 47.29 | 70.64 | 16.48/15.91 | 33.11 |

- For BERT on GLUE benchmark, the accuracy degradation is usually within 0—0.2 for W8A8 and about 1–2 (except RTE for BERT$_{large}$) for W4/8A8 (Table 2, 3, E.1, and E.2).

- For GPT-3-style models (350M and 1.3B) with W8A8, the average accuracy degradation is about 0.1% and the perplexity drop is about 0.3 points. However, for each of individual tasks, due to the nature of zero-shot evaluation (no task-specific fine-tuning, different number of eval cases between tasks, etc), the accuracy change varies. Particularly, for some cases, W8A8 even outperforms FP16.

- With W4/8A8, (a) for the 350M GPT-3-style model, the average accuracy degradation is about 2.3% and the perplexity drop is about 9.5; (b) for the 1.3B model, the average accuracy degradation is 2.5% and the perplexity drop is about 2.5. Note that those results are achieved without tuning the layer-by-layer knowledge distillation.

- For GPT-J (6B), our INT8 model has about 0.3 average perplexity loss; and for GPT-NeoX (20B), our INT8 model has about 0.08% accuracy loss.

## C   PTQ challenge of BERT$_{base}$

From Table C.1, we observe similar results as [7], where the accuracy degradation of INT8 quantization is mainly from activation quantization. Specifically, there is a negligible accuracy drop from INT8 weight quantization (i.e., the row of W8A16). However, with sole INT8 activation (i.e., the row of W16A8), the accuracy decreases from 84.06 to 79.61. Besides, we also push the weight quantization to a mixed-precision setting with INT4 for weights in FFC and INT8 for weights in MHSA (i.e., the row of W4/8A16). This ultra-low precision quantization leads the model to be purely random without meaning prediction.

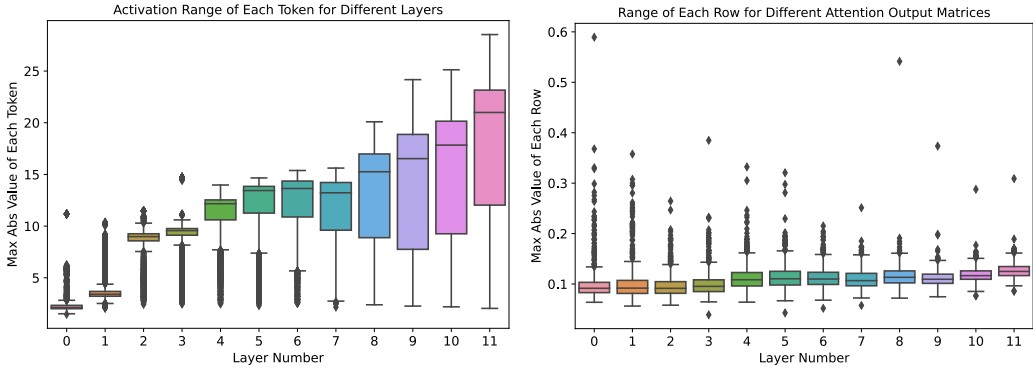

Figure C.1: The activation range of different layers (left) and the row-wise weight range of the attention output matrix ($\boldsymbol{W}_o$) of different layers (right). The results are based on the BERT$_{base}$ trained on MNLI dataset. Please see Figure 2 for the results of GPT-3$_{350M}$.

# D Details about System Optimization

By having the weight and activation quantization, we can use the GeMM schedule that exploits the INT8 Tensor-core units which provide 2x/4x more compute efficiency compared to the FP16/FP32 Tensor cores. For this purpose, we adapt the CUTLASS library to produce multiple schedules based on the input sizes we are considering in our application, such as the batch size, sequence length, and the Transformer hidden dimension. To achieve the best latency, we also develop our own efficient parallel implementation of the quantization operator on GPU. During the inference run-time, based on the total batch size ($batch \times seq_len$), we choose the schedule that results in the lowest possible padding when performing the Tensor-core matrix-multiplication operations.

To find the best schedule for the GeMM operation, we use the CUTLASS profiler tool that explores the tiling dimensions on the thread-blocks, WARPs, and WMMA (Tensor cores), as the three compute hierarchies available within the Ampere GPU architecture. Then, we find the best schedule by sorting the tile-based schedule based on either peak throughput achieved on the large-batch case, or the maximum memory bandwidth taken from the main memory when the batch size is small.

However, there are still several challenges we need to address which are discussed below.

**Operation Fusion for Token-wise Activation Quantization.** One of the main challenges of our quantization scheme is how to efficiently quantize hidden states before the GeMM operation. In order to remove the overhead, we fuse the activation quantization with its associated element-wise and/or reduction-based operations such as bias-addition, GELY, and LayerNorm. This is due to the fact that each SM takes care of one row (token) of the activation and therefore, we can reuse the computation from the thread registers and compute the quantization scale, avoiding the data movement between GPU kernels and main memory. Moreover, by converting data from FP16 to INT8, we can utilize the memory bandwidth twice, further improving the inference latency and throughput.

**Dequantization Associated with GeMM Schedule** To utilize the output of integer output from GeMM operator in the following operators, one important step is to dequantize the output by using the scaling factor of the weight and activations. This dequantization step generally introduces extra overhead for quantized network inference due to the data movement. As such, we add a custom epilogue, which converts the final accumulated result (from INT32 format) of each row and column of the output to the real value (in FP16 format), using corresponding floating-point quantization scales computed from weight and activation group-wise quantization. By fusing the dequantization with GeMM schedule, we ensure that there is no overhead exposed by using the INT8 operations while producing the FP16 results that are used in the following operation.

Furthermore, to effectively combine dequantization with the GeMM operation, we read the two groups of quantization scales for the activation and weight matrices in advance prior to completion of the multiplication of the output matrix. Doing so, we overlap the reading of the extra quantization parameters with the GeMM computation and the GeMM-plus-dequantization can seamlessly work together without stalling the inference pipeline.

**Cuda Graph Enhanced Small Model Inference.** As the execution time for specific kernels reduce by optimizing the throughput using the INT8 inference pipeline, the overhead of launching the GPU kernels and the CPU-to-GPU communication become a major bottleneck mostly on small-scale models. To address this issue, we add the CUDA-Graph support to our inference pipeline that reduces the CPU overhead, by storing the trace of the kernels launched during the inference forward computation, and creating the computation-graph to be reused in the next call to the inference pipeline. Thus, after storing the graph for the first time, we can replay the graph for the following requests, which substantially improves the performance especially on small models, such as BERT$_{base}$. For a fair comparison, we also enable Cuda Graph for FP16 baseline.

**Other Operator Fusions** Besides quantization/dequantization operator fusion, other fusion methods, which fuse element-wise operations, matrix multiplications, transpositions, and reductions all into a single kernel, are also applied. These can significantly reduce the number of kernel invocations as well as main memory access to reduce the main memory access latency. Please see [2] for more details.

Table E.1: Result of BERT$_{base}$ on the development set of GLUE benchmark (except WNLI). Here WxAy means x-bit for weight quantization and y-bit for activation quantization. Particularly, for W4/8, we quantize the MHSA's weight to INT8 and FFC's weight to INT4. Please see Appendix A.3 for the reported metrics.

| Precision (Method) | CoLA | MNLI-m | MNLI-mm | MRPC | QNLI | QQP | RTE | SST-2 | STS-B | Ave. |
|---|---|---|---|---|---|---|---|---|---|---|
| W16A16 (Baseline) | 59.72 | 84.94 | 85.06 | 86.27/90.57 | 92.15 | 91.51/88.56 | 72.20 | 93.23 | 90.06/89.59 | 83.95 |
| W8A8 (ZeroQuant-LKD No Tuning) | 59.59 | 84.83 | 85.13 | 86.03/90.39 | 91.98 | 91.45/88.46 | 71.12 | 93.12 | 90.09/89.62 | 83.75 |
| W8A8 (ZeroQuant-LKD Tuned) | 60.90 | 84.95 | 85.10 | 86.27/90.60 | 92.07 | 91.47/88.47 | 71.84 | 93.46 | 90.09/89.62 | 84.07 |
| W4/8A16 (ZeroQuant-LKD No Tuning) | 58.50 | 83.16 | 83.69 | 84.80/89.31 | 90.83 | 88.94/84.12 | 70.04 | 92.78 | 88.49/88.67 | 82.35 |
| W4/8A16 (ZeroQuant-LKD Tuned) | 60.04 | 83.64 | 84.31 | 85.78/89.53 | 91.01 | 90.66/87.26 | 71.84 | 93.12 | 88.68/88.79 | 83.26 |
| W4/8A8 (ZeroQuant-LKD No Tuning) | 58.80 | 83.09 | 83.65 | 85.78/89.90 | 90.76 | 89.32/84.85 | 71.84 | 93.00 | 88.16/88.55 | 82.71 |
| W4/8A8 (ZeroQuant-LKD Tuned) | 60.30 | 83.47 | 84.03 | 85.78/89.90 | 90.87 | 90.77/87.38 | 71.84 | 93.00 | 88.38/88.70 | 83.22 |

Table E.2: Result of BERT$_{large}$ on the development set of GLUE benchmark (except WNLI). Here WxAy means x-bit for weight quantization and y-bit for activation quantization. Particularly, for W4/8, we quantize the MHSA's weight to INT8 and FFC's weight to INT4. Please see Appendix A.3 for the reported metrics.

| Precision (Method) | CoLA | MNLI-m | MNLI-mm | MRPC | QNLI | QQP | RTE | SST-2 | STS-B | Ave. |
|---|---|---|---|---|---|---|---|---|---|---|
| W16A16 (Baseline) | 63.35 | 86.65 | 85.91 | 87.99/91.62 | 92.24 | 91.08/88.08 | 74.01 | 93.46 | 90.34/90.11 | 85.03 |
| W8A8 (ZeroQuant-LKD No Tuning) | 63.38 | 86.52 | 85.64 | 87.75/91.50 | 92.31 | 91.09/88.05 | 72.56 | 93.35 | 90.45/90.19 | 84.81 |
| W8A8 (ZeroQuant-LKD Tuned) | 64.36 | 86.64 | 85.74 | 88.48/91.97 | 92.49 | 91.15/88.13 | 74.73 | 93.58 | 90.45/90.19 | 85.30 |
| W4/8A16 (ZeroQuant-LKD No Tuning) | 63.72 | 84.90 | 84.81 | 87.99/91.39 | 91.45 | 90.34/86.92 | 51.62 | 92.43 | 89.46/89.29 | 81.85 |
| W4/8A16 (ZeroQuant-LKD Tuned) | 64.06 | 85.02 | 84.98 | 88.73/91.99 | 91.82 | 90.45/87.12 | 52.35 | 92.78 | 89.72/89.44 | 82.19 |
| W4/8A8 (ZeroQuant-LKD No Tuning) | 63.51 | 84.70 | 84.71 | 88.73/91.99 | 91.73 | 90.25/86.74 | 49.82 | 92.09 | 89.34/89.08 | 81.62 |
| W4/8A8 (ZeroQuant-LKD Tuned) | 63.60 | 84.77 | 84.90 | 88.97/92.15 | 91.87 | 90.37/86.99 | 50.54 | 92.55 | 89.57/89.38 | 81.88 |

# E  Tuned Results on BERT

As mentioned in the main text and Appendix A.2, we use the same set of hyperparameters for BERT. However, tuning them can significantly boost the performance for ZeroQuant. Here, we tune two hyperparameters, i.e., the learning rate and the number of iterations in order to show the best possible performance of ZeroQuant on both BERT$_{base}$ and BERT$_{large}$. Particularly, we choose learning rate from the set {1e-6, 2e-6, 5e-6, 1e-5}, and choose number of iterations from the set {0, 50, 100, 200, 400, 800, 1600}. Thanks to the lightweight of LKD, the total tuning time for BERT$_{base}$ (including all data loading time, evaluation time, tokenization time, all three quantization schemes, etc) is around 4.5 hours on 8 40G-A100 GPUs (i.e., 36 GPU hours), and the tuning time for BERT$_{large}$ is around 16 hours on 8 40G-A100 GPUs (i.e., 128 GPU hours).

We summarize the best results in the Table E.1 and E.2.

# F  QAT on BERT$_{large}$

We use four different learning rates for QAT on BERT$_{large}$, {5e-6, 1e-5, 2e-5, 5e-5}. The final results we reported in the paper are chosen from the best single run among those four different learning rates. However, even with such tuning, we are not able to get good performance for BERT$_{large}$ on RTE.

Also, note that the time cost we used in the main text is based on a single run. if we consider the tuning cost, the total time will be $4 \times 7181s$

# G  Other Results

## G.1  TinyBERT

Our proposed method can also be applied to small Transformer models as well. One example is BERT$_{base}$ used in the main text. To further demonstrate this, we use TinyBERT (4L-312H[5]) with ZeroQuant W8A8 (without LKD) as an example (The checkpoint of CoLA is broken so we did not include the result of CoLA). The results are shown in Table G.1. As can be seen, the performance of ZeroQuant (INT8) is comparable with the original FP16 baseline.

---

[5] https://huggingface.co/Sayan01

Table G.1: Result of TinyBERT on the development set of GLUE benchmark (except WNLI). Here WxAy means x-bit for weight quantization and y-bit for activation quantization. Please see Appendix A.3 for the reported metrics.

| Precision (Method) | CoLA | MNLI-m | MNLI-mm | MRPC | QNLI | QQP | RTE | SST-2 | STS-B |
|---|---|---|---|---|---|---|---|---|---|
| W16A16 (Baseline) | – | 80.1 | 80.2 | 83.3/88.8 | 83.7 | 88.5/85.1 | 66.8 | 87.0 | 87.3/87.1 |
| W8A8 (ZeroQuant) | – | 80.1 | 79.9 | 82.1/88.2 | 84.1 | 88.7/85.2 | 67.5 | 86.9 | 87.6/87.4 |

Table G.2: Results of ZeroQuant on ResNet family. we report both Top-1 and Top-5 accuracy here.

| Precision (Method) | ResNet18 | ResNet50 | ResNet152 |
|---|---|---|---|
| W16A16 (Baseline) | 69.766 / 89.068 | 76.132 / 92.862 | 78.318 / 94.050 |
| W8A8 (ZeroQuant) | 69.686 / 89.004 | 76.022 / 92.848 | 78.272 / 94.002 |

## G.2 CNN Results

We can still apply our quantization scheme on convolutional neural networks with some modification. For instance, we can use per-output-channel quantization for convolutional kernels as well as per-image quantization for hidden states (images). As such, this quantization is still satisfied with the hardware compute constraint since we do not break the computation granularity of GPUs (i.e., real speedup can still be achieved on GPUs). To verify this, we applied this scheme for ResNet18/ResNet50/ResNet152 with Top-1 and Top-5 accuracy. The code base is from PyTorch Example[6]. As can be seen from Table G.2, the accuracy degradation of ZeroQuant is 0.1% as compared to FP32 models.

## G.3 ViT Result

Our method can be applied to ViTs as well. We chose Huggingface's ViT checkpoint finetuned on ImageNet-1K to test ZeroQuant (INT8) and PTQ. We provide both Top-1 and Top-5 accuracy in Table G.3.

## G.4 Comparison to Per-column Quantization

As requested by review, we add the comparison between ZeroQuant and per-column quantization used in [74]. First of all, we want to clarify that the authors of [74] mainly focus on the algorithm accuracy evaluation, and they did not provide real speedup results. However, our paper focuses on both algorithm and hardware constraint/implementation, and we provide real speedup solutions.

Here, we include the comparison of ZeroQuant and the work of [74] (use per-column for weight and per-row for tokens) in Table G.4. The accuracy of ZeroQuant and the method from [74]. is similar to each other.

## G.5 Comparison among Different Quantization Schemes

Asymmetric quantization introduces the zero-point (or bias term), which will affect the inference speed performance (this bias-term introduces the extra matrix-vector production). That's the reason why in ZeroQuant we choose symmetric quantization as the quantization scheme. We did not directly provide asymmetric quantization but MP-PTQ in Table 2 applied asymmetric activation quantization. To further demonstrate that asymmetric quantization cannot fully resolve the outlier issue (and with higher cost due to the extra matrix-vector production), we provided asymmetric quantization results for PTQ (W8A8) on $BERT_{base}$ and $BERT_{large}$ in Table G.5 and G.6.

As can be seen, asymmetric quantization boosts the PTQ accuracy a lot but it is still >1 point lower than FP16 BERT-base. However, our ZeroQuant can close this gap to 0.25 without involving any extra

---

[6]https://github.com/pytorch/examples/blob/main/imagenet/main.py

Table G.3: Results of ZeroQuant on ViT. we report both Top-1 and Top-5 accuracy here.

| Precision (Method) | ViT-base |
|---|---|
| W16A16 (Baseline) | 81.430 / 96.010 |
| PTQ | 80.110 / 95.318 |
| W8A8 (ZeroQuant) | 81.464 / 95.988 |

Table G.4: Comparison between ZeroQuant and [74]. Please see Appendix A.3 for the reported metrics.

| Precision (Method) | CoLA | MNLI-m | MNLI-mm | MRPC | QNLI | QQP | RTE | SST-2 | STS-B | Ave. |
|---|---|---|---|---|---|---|---|---|---|---|
| W16A16 (Baseline) | 59.72 | 84.94 | 85.06 | 86.27/90.57 | 92.15 | 91.51/88.56 | 72.20 | 93.23 | 90.06/89.59 | 83.95 |
| W8A8 [74] | 59.62 | 84.86 | 85.09 | 85.78/90.20 | 91.93 | 91.48/88.48 | 71.12 | 93.23 | 90.02/89.53 | 83.73 |
| W8A8 (ZeroQuant) | 59.59 | 84.83 | 85.13 | 86.03/90.39 | 91.98 | 91.45/88.46 | 71.12 | 93.12 | 90.09/89.62 | 83.75 |

matrix-vector production for the bias/zero-point term from asymmetric quantization. For BERT$_{large}$, asymmetric PTQ has 2 points accuracy degradation and ZeroQuant only has 0.22.

## G.6 Five-seed Results

First, we want to clarify that all the results we reported in the main text do not use any hyper-parameter tuning. For W8A8, since there is no training involved, there is no tuning at all. Also, for W4/8A16 or W4/8A8, the main text results use a fixed set of hyperparameters, which means there is no hyper-parameter tuning. This part is mentioned with the detailed setting in Appendix A. The tuning results in Appendix E are used to demonstrate the low-cost of layer-by-layer knowledge distillation as well as for better comparison if someone applied hyperparameter search.

For now, the results we used in the paper are based on a single seed. The main reason is that layer-by-layer knowledge distillation is a lightweight training procedure. As such, the weight won't be significantly changed during LKD finetuning. To further alleviate the concern about variance/noise, we run the results for BERT-base with W4/8A16 and W4/8A8 with 5 seeds and report the results here. We follow the same setting as we used in the main text. Particularly, for LKD, we use 100 iterations with batch size 32 and sequence length 128, and we fix the learning rate as 5e-6. For group-wise quantization, we choose 48 groups for all weight matrices. We use seed {40, 41, 42, 43, 44} for this experiment. The results are shown in Table G.7. As can be seen, the 5-seed average result is similar to what we have reported in the main text.

## G.7 INT8 Comparison between cuBLAS and CUTLASS

For the INT8 kernels, we run a profiling on different tiling schedules for thread-blocks, WARPs, and tensor-cores (WMMA instructions) in order to get the best latency based on the model dimension and batch size. Note that there are some paddings needed to keep the tenor shapes 16, 32, 64, or 128-aligned in order to maximize the performance when configuring the CUTLASS GeMM kernels. In our inference system, we choose the best scheduling for each application that minimizes the padding and gives the best performance. In our experiments, we see much better performance on the INT8 kernels for CUTLASS vs cuBLAS. Furthermore, we can fuse the dequantization step by modifying the CUTLASS kernels whereas this is not an option for cuBLAS library as it is not open-sourced.

We included detailed profiling results in Table G.8 for BERT-scale matrix sizes and GPT-J/GPT-NeoX-scale matrix sizes. The hardware used is the 40GB-A100. Here we use bsz to represent batch size, and seq to represent sequence length. The time is computed based on 1000 runs average time and the TFLOPS here means the Tera-Flops/s the hardware achieved. As we can see, our highly optimized CUTLASS INT8 GEMM is much faster than FP16/INT8 CUBLAS GEMM.

Table G.5: Comparison between symmetric and asymmetric quantization schemes on BERT$_{base}$. Please see Appendix A.3 for the reported metrics.

| Precision (Method) | CoLA | MNLI-m | MNLI-mm | MRPC | QNLI | QQP | RTE | SST-2 | STS-B | Ave. |
|---|---|---|---|---|---|---|---|---|---|---|
| W16A16 (Baseline) | 59.72 | 84.94 | 85.06 | 86.27/90.57 | 92.15 | 91.51/88.56 | 72.20 | 93.23 | 90.06/89.59 | 83.95 |
| W8A8 (PTQ, Symmetric) | 56.06 | 79.99 | 81.06 | 75.49/79.67 | 87.35 | 89.92/86.82 | 48.38 | 91.40 | 86.58/86.44 | 77.41 |
| W8A8 (PTQ, Asymmetric) | 59.94 | 80.58 | 81.54 | 84.80/89.67 | 92.00 | 91.42/88.30 | 71.12 | 92.89 | 89.34/89.30 | 82.72 |
| W8A8 (ZeroQuant) | 59.59 | 84.83 | 85.13 | 86.03/90.39 | 91.98 | 91.45/88.46 | 71.12 | 93.12 | 90.09/89.62 | 83.75 |

Table G.6: Comparison between symmetric and asymmetric quantization schemes on BERT$_{large}$. Please see Appendix A.3 for the reported metrics.

| Precision (Method) | CoLA | MNLI-m | MNLI-mm | MRPC | QNLI | QQP | RTE | SST-2 | STS-B | Ave. |
|---|---|---|---|---|---|---|---|---|---|---|
| W16A16 (Baseline) | 63.35 | 86.65 | 85.91 | 87.99/91.62 | 92.24 | 91.08/88.08 | 74.01 | 93.46 | 90.34/90.11 | 85.03 |
| W8A8 (PTQ, Symmetric) | 60.57 | 75.69 | 76.94 | 81.13/84.93 | 88.49 | 84.04/74.35 | 46.93 | 91.74 | 62.57/55.77 | 73.54 |
| W8A8 (PTQ, Asymmetric) | 60.52 | 86.16 | 85.84 | 85.29/89.40 | 92.11 | 90.09/86.16 | 64.98 | 92.89 | 89.40/89.38 | 83.04 |
| W8A8 (ZeroQuant) | 63.38 | 86.52 | 85.64 | 87.75/91.50 | 92.31 | 91.09/88.05 | 72.56 | 93.35 | 90.45/90.19 | 84.81 |

## G.8 Effectiveness of Group Quantization and Token Quantization

We summarize the results in Table G.9. Besides directly adopting group quantization (GQ) from Q-BERT, we also include the real GPU kernel implementation. Without the GPU kernel part, there is no real speedup benefit for group quantization. Meanwhile, without token-wise quantization (TQ), the average accuracy of GQ is 66.52. TQ can further bring this number to 81.06 (14.54 accuracy gain, which is non-trivial). This demonstrates the benefit of TQ.

## G.9 Relative Error Analysis

We use the relative error results for BERT$_{base}$ of the quantized models to support the claim that the proposed method can deal with the dynamic range of full-precision activations and weights. Particularly, the relative error is computed as

$$Average(\frac{|A_{quantize} - A_{full}|}{|A_{full}| + 1e - 8}, \tag{3}$$

where A means the weight or activation, and the subscript quantize (full) means the quantized (FP16) version.

The results are summarized in Table G.10. The proposed methods in ZeroQuant can significantly reduce the relative error as compared to the standard quantization method.

## G.10 Comparison with MP-PTQ and Q-BERT of BERT$_{large}$

Since the original Q-BERT and MP-PTQ do not include BERT-large study, we implement our own code to produce the result of Q-BERT based on the paper's description and use the open-sourced code[7] to produce the result of MP-PTQ (we use –quant-dict "'y': 16, 'h': 16, 'x': 16, 'P': 16, 'C': 16" and –quant-setup MSE_logits for STS-B task, and we use –quant-dict "'y': 16, 'h': 16, 'x': 16" for other tasks). The results are shown in Table G.11. As can be seen, ZeroQuant achieves similar accuracy as Q-BERT but without any training cost.

For MP-PTQ, we can reproduce the paper's result for BERT-base models. However, for BERT-large, all tasks give random guess except CoLA (we tried three different kinds of checkpoints, including Huggingface's checkpoint, our own checkpoints, and checkpoints trained using the MP-PTQ repo). As such, we did not include the rest of the results. After increasing the bit-precision for activation from 8 to 10 bits, the accuracy of MP-PTQ is significantly boosted. Therefore, the quantization error of activation is the main cause of MP-PTQ on BERT-large and this is primarily from the dynamic range of activations as what we discussed in Section C. Also, this result (i.e., MP-PTQ works for BERT-base but not BERT-large out of the box) is aligned with our reported PTQ results in Table 2 and 3, i.e., BERT-large's W8A8 has lower accuracy than BERT-base's W8A8, which means BERT-large is more sensitive to dynamic activation ranges.

---

[7] https://github.com/qualcomm-ai-research/transformer-quantization

Table G.7: Five-seed results on BERT$_{base}$. Please see Appendix A.3 for the reported metrics.

| Precision (Method) | CoLA | MNLI-m | MNLI-mm | MRPC | QNLI | QQP | RTE | SST-2 | STS-B | Ave. |
|---|---|---|---|---|---|---|---|---|---|---|
| W16A16 (Baseline) | 59.72 | 84.94 | 85.06 | 86.27/90.57 | 92.15 | 91.51/88.56 | 72.20 | 93.23 | 90.06/89.59 | 83.95 |
| W4/8A16 (Single Seed) | 58.50 | 83.16 | 83.69 | 84.80/89.31 | 90.83 | 88.94/84.12 | 70.04 | 92.78 | 88.49/88.67 | 82.35 |
| W4/8A16 (Five Seeds) | 58.64 | 83.15 | 83.74 | 84.65/89.22 | 90.87 | 89.08/84.39 | 70.40 | 92.82 | 88.48/88.65 | 82.43 |
| W4/8A8 (Single Seed) | 58.80 | 83.09 | 83.65 | 85.78/89.90 | 90.76 | 89.16/84.85 | 71.84 | 93.00 | 88.16/88.55 | 82.71 |
| W4/8A8 (Five Seeds) | 58.76 | 83.03 | 83.63 | 84.75/89.26 | 90.78 | 89.45/85.97 | 71.12 | 92.78 | 88.16/88.56 | 82.52 |

Table G.8: GeMM Speed Comparison.

| BERT | | FP16 CUBLAS | | | INT8 cuBLAS | | INT8 CUTLASS | |
|---|---|---|---|---|---|---|---|---|
| bsz | seq | Matrix dim | time (ms) | TFLOPS | time (ms) | TFLOPS | time (ms) | TFLOPS |
| 1 | 128 | 768 | 0.02355 | 6.411796 | 0.021473 | 7.031874 | 0.009805 | 15.39947 |
| 1 | 128 | 1024 | 0.019223 | 13.9643 | 0.022616 | 11.86932 | 0.010406 | 25.79559 |
| 1 | 256 | 768 | 0.018297 | 16.50514 | 0.025432 | 11.87458 | 0.009844 | 30.67661 |
| 1 | 256 | 1024 | 0.019206 | 27.95322 | 0.060184 | 8.920457 | 0.010737 | 50.00222 |
| 64 | 128 | 768 | 0.049335 | 195.8768 | 0.153681 | 62.88129 | 0.035627 | 271.2449 |
| 64 | 128 | 1024 | 0.077738 | 220.9962 | 0.255213 | 67.31586 | 0.050085 | 343.0171 |
| 64 | 256 | 768 | 0.096026 | 201.2717 | 0.303941 | 63.58921 | 0.064837 | 298.0926 |
| 64 | 256 | 1024 | 0.15432 | 222.6525 | 0.500961 | 68.58771 | 0.092166 | 372.8039 |
| GPT | | | | | | | | |
| bsz | seq | Matrix dim | time (ms) | TFLOPS | time (ms) | TFLOPS | time (ms) | TFLOPS |
| 1 | 1 | 4096 | 0.027091 | 1.23857 | 0.031635 | 1.060677 | 0.017066 | 1.966157 |
| 1 | 1 | 6144 | 0.066214 | 1.140199 | 0.073625 | 1.025434 | 0.036825 | 2.050146 |
| 1 | 8 | 4096 | 0.025938 | 10.34901 | 0.061419 | 4.37056 | 0.017056 | 15.73871 |
| 1 | 8 | 6144 | 0.064703 | 9.334616 | 0.175115 | 3.44904 | 0.036513 | 16.54146 |
| 1 | 16 | 4096 | 0.026499 | 20.25984 | 0.061636 | 8.710351 | 0.017051 | 31.48534 |
| 1 | 16 | 6144 | 0.065374 | 18.47757 | 0.175221 | 6.893921 | 0.036419 | 33.16868 |

## G.11 Different loss choices of LKD

We tried both L2 and L1 losses, and the results from those two losses are very similar. Below we provide the results of W4/8A8 on BERT-base. As can be seen from Table G.12, the accuracy of L1 loss and L2 loss based LKD is similar to each other.

## H Limitations and Future Work

We believe it is critical for every work to clearly state its limitations, especially in this area. One limitation is that in this work we only focused on natural language models, but it would be interesting to see how ZeroQuant would perform for computer vision models. We leave this as a future work.

Another limitation is that we can only verify the scalability of ZeroQuant up to 20B scale models. If there are new releases of larger open-sourced models, it would be great to test ZeroQuant on those larger models as well.

Third, in this paper, we found out that the activation input of self-attention is more sensitive for quantization for the extra-large model (GPT-NeoX$_{20B}$). However, we are unable to verify this on other extra-large models due to the lack of open-sourced models.

## I Full Zero-shot Evaluation of GPT-3-style Models

We includes all zero-shot evaluation results in this section for all GPT-3-style models, inlcuding GPT-NeoX$_{20B}$.

Table G.9: Results of effectiveness of group quantization and token-wise quantization. Please see Appendix A.3 for the reported metrics.

| Precision | GQ | TQ | CoLA | MNLI-m | MNLI-mm | MRPC | QNLI | QQP | RTE | SST-2 | STS-B | Ave. |
|---|---|---|---|---|---|---|---|---|---|---|---|---|
| W16A16 | – | – | 59.72 | 84.94 | 85.06 | 86.27/90.57 | 92.15 | 91.51/88.56 | 72.20 | 93.23 | 90.06/89.59 | 83.95 |
| W8A8 | No | No | 56.06 | 79.99 | 81.06 | 75.49/79.67 | 87.35 | 89.92/86.82 | 48.38 | 92.66 | 86.58/86.44 | 77.41 |
| W8A8 | Yes | No | 59.84 | 80.25 | 81.37 | 83.82/89.18 | 91.23 | 91.32/88.14 | 70.04 | 93.12 | 88.66/88.80 | 82.31 |
| W8A8 | Yes | Yes | 59.59 | 84.83 | 85.13 | 86.03/90.39 | 91.98 | 91.45/88.46 | 71.12 | 93.12 | 90.09/89.62 | 83.75 |

Table G.10: Results of relative error of group quantization and token-wise quantization as compared to standard quantization methods. Please see Appendix A.3 for the reported metrics.

| | Standard Quantization | GQ or TQ |
|---|---|---|
| Weight, INT8 | 0.10 | 0.05 |
| Weight, INT4 | 0.73 | 0.43 |
| Activation, INT8 | 0.22 | 0.13 |

Table G.11: Comparison between ZeroQuant and Q-BERT/MT-PTQ on BERT$_{large}$. Please see Appendix A.3 for the reported metrics.

| Precision (Method) | CoLA | MNLI-m | MNLI-mm | MRPC | QNLI | QQP | RTE | SST-2 | STS-B | Ave. | Time |
|---|---|---|---|---|---|---|---|---|---|---|---|
| W16A16 (Baseline) | 63.35 | 86.65 | 85.91 | 87.99/91.62 | 92.24 | 91.08/88.08 | 74.01 | 93.46 | 90.34/90.11 | 85.03 | – |
| W8A8 (Q-BERT) | 62.86 | 86.26 | 86.43 | 86.27/90.38 | 92.07 | 91.59/88.66 | 73.73 | 93.92 | 89.87/89.54 | 84.83 | 718 |
| W8A8/16 (MT-PTQ) | 61.48 | – | – | – | – | – | – | – | – | – | – |
| W8A10/16 (MT-PTQ) | 62.6 | 85.68 | 85.68 | 87.25/91.22 | 91.37 | 90.51/87.09 | 72.56 | 93.00 | 85.2/87.07 | 83.90 | 10 |
| W8A8 (ZeroQuant) | 63.38 | 86.52 | 85.64 | 87.75/91.50 | 92.31 | 91.09/88.05 | 72.56 | 93.35 | 90.45/90.19 | 84.81 | 0 |

Table G.12: Loss choices of LKD on BERT$_{base}$. Please see Appendix A.3 for the reported metrics.

| Precision (Method) | CoLA | MNLI-m | MNLI-mm | MRPC | QNLI | QQP | RTE | SST-2 | STS-B | Ave. |
|---|---|---|---|---|---|---|---|---|---|---|
| W16A16 (Baseline) | 59.72 | 84.94 | 85.06 | 86.27/90.57 | 92.15 | 91.51/88.56 | 72.20 | 93.23 | 90.06/89.59 | 83.95 |
| W4/8A8 (L1) | 58.90 | 83.41 | 83.84 | 83.82/89.00 | 90.55 | 88.92/84.12 | 73.29 | 92.66 | 88.18/88.56 | 82.66 |
| W4/8A8 (L2) | 58.80 | 83.09 | 83.65 | 85.78/89.90 | 90.76 | 89.16/84.85 | 71.84 | 93.00 | 88.16/88.55 | 82.71 |

Table I.1: The full results of GPT-3$_{350M}$.

| Tasks | Baseline | PTQ | | | | ZeroQuant | | | ZeroQuant-LKD | |
|---|---|---|---|---|---|---|---|---|---|---|
| | W16A16 | W8A16 | W16A8 | W8A8 | W4/8A16 | W8A8 | W4/8A16 | W4/8A8 | W4/8A16 | W4/8A8 |
| HellaSwag | 38.6 | 38.1 | 37.6 | 36.8 | 26.5 | 38.4 | 30.4 | 30.5 | 35.3 | 35.3 |
| LAMBADA | 49.3 | 49.3 | 44.7 | 42.9 | 0 | 51.0 | 10.1 | 10.5 | 39.8 | 37.4 |
| TriviaQA | 3.00 | 2.67 | 2.70 | 2.32 | 0 | 2.86 | 0.159 | 0.194 | 1.043 | 0.23 |
| WebQs | 1.43 | 0.935 | 1.23 | 0.689 | 0 | 1.378 | 0.246 | 0.394 | 0.591 | 0.049 |
| Winogrande | 53.2 | 52.1 | 52.1 | 52.1 | 47.8 | 51.4 | 52.6 | 50.7 | 51.6 | 51.8 |
| PIQA | 66.3 | 66.1 | 64.8 | 64.1 | 51.4 | 66.5 | 58.5 | 57.7 | 63.8 | 61.8 |
| ARC (Challenge) | 24.2 | 24.0 | 24.0 | 24.1 | 27.0 | 24.5 | 22.0 | 21.8 | 21.8 | 23.6 |
| ARC (Easy) | 45.5 | 44.7 | 44.2 | 43.9 | 25.1 | 44.5 | 37.6 | 37.5 | 40.5 | 40.5 |
| ANLI R1 | 31.1 | 30.0 | 31.3 | 33.2 | 33.4 | 31.1 | 32.8 | 32.7 | 32.4 | 33.8 |
| ANLI R2 | 34.3 | 36.0 | 36.5 | 35.9 | 33.4 | 34.3 | 34.7 | 34.2 | 34.1 | 33.5 |
| ANLI R3 | 34.1 | 34.0 | 33.0 | 37.2 | 33.5 | 33.4 | 34.9 | 34.5 | 33.1 | 33.4 |
| OpenBookQA | 29.4 | 29.6 | 28.2 | 28.0 | 30.2 | 29.2 | 27.2 | 28.0 | 29.4 | 28.2 |
| RACE-h | 32.4 | 31.3 | 30.3 | 30.7 | 22.4 | 32.2 | 25.7 | 26.4 | 29.5 | 29.7 |
| BoolQ | 60.3 | 60.2 | 57.0 | 56.9 | 37.8 | 60.2 | 60.1 | 59.4 | 61.9 | 61.9 |
| Copa | 69.0 | 67.0 | 71.0 | 73.0 | 48.0 | 69.0 | 63.0 | 64.0 | 68.0 | 66.0 |
| RTE | 53.8 | 54.2 | 52.7 | 53.1 | 52.7 | 53.4 | 52.0 | 52.7 | 53.1 | 53.1 |
| WSC | 36.5 | 36.5 | 36.5 | 35.6 | 63.5 | 36.5 | 36.5 | 36.5 | 36.5 | 36.5 |
| MultiRC | 0.839 | 0.839 | 0.839 | 0.944 | 0.315 | 0.839 | 1.889 | 1.889 | 0.839 | 0.839 |
| ReCoRD | 75.1 | 74.8 | 69.2 | 67.5 | 16.1 | 74.9 | 56.5 | 55.3 | 70.1 | 68.5 |
| Wikitext-2 | 21.52 | 22.09 | 24.56 | 26.20 | 1.76e5 | 21.68 | 88.64 | 92.10 | 30.56 | 31.13 |
| Average Acc | 38.86 | 38.54 | 37.78 | 37.84 | 28.9 | 38.71 | 33.52 | 33.42 | 37.02 | 36.64 |

Table I.2: The full results of GPT-3$_{1.3B}$.

| Tasks | Baseline | PTQ | | ZeroQuant | | | ZeroQuant-LKD | |
|---|---|---|---|---|---|---|---|---|
| | W16A16 | W8A8 | W4/8A16 | W8A8 | W4/8A16 | W4/8A8 | W4/8A16 | W4/8A8 |
| HellaSwag | 51.4 | 47.0 | 26.1 | 50.8 | 43.7 | 43.2 | 48.5 | 46.7 |
| LAMBADA | 61.3 | 54.8 | 0 | 62.6 | 43.9 | 46.8 | 59.4 | 48.7 |
| TriviaQA | 7.37 | 4.43 | 0 | 6.67 | 2.36 | 2.09 | 4.28 | 2.99 |
| WebQs | 2.90 | 1.476 | 0 | 2.07 | 1.132 | 1.28 | 1.673 | 1.083 |
| Winogrande | 57.1 | 55.7 | 50.1 | 57.1 | 54.6 | 54.3 | 55.3 | 53.8 |
| PIQA | 71.4 | 67.7 | 50.4 | 70.7 | 66.5 | 66.4 | 69.5 | 68.1 |
| ARC (Challenge) | 27.2 | 27.1 | 26.5 | 26.8 | 25.7 | 25.3 | 27.8 | 26.5 |
| ARC (Easy) | 54.5 | 49.7 | 26.0 | 53.8 | 48.0 | 47.0 | 52.2 | 50.3 |
| ANLI R1 | 32.0 | 33.1 | 33.0 | 33.4 | 33.8 | 33.6 | 34.2 | 33.8 |
| ANLI R2 | 32.0 | 32.9 | 33.3 | 33.9 | 33.0 | 33.0 | 33.8 | 32.8 |
| ANLI R3 | 33.8 | 33.5 | 32.3 | 34.8 | 33.6 | 33.5 | 33.7 | 33.0 |
| OpenBookQA | 33.6 | 32.6 | 27.0 | 33.4 | 30.0 | 28.8 | 31.6 | 29.0 |
| RACE-h | 33.6 | 32.6 | 22.4 | 32.7 | 30.9 | 29.9 | 32.7 | 33.2 |
| BoolQ | 62.4 | 59.2 | 37.8 | 61.3 | 60.3 | 59.8 | 61.7 | 61.3 |
| Copa | 70.0 | 70.0 | 55.0 | 72.0 | 73.0 | 74.0 | 72.0 | 70.0 |
| RTE | 53.1 | 54.5 | 50.9 | 52.7 | 52.7 | 52.7 | 52.7 | 52.0 |
| WSC | 37.5 | 36.5 | 63.5 | 36.5 | 36.5 | 36.5 | 36.5 | 36.5 |
| MultiRC | 1.05 | 0.839 | 0.315 | 0.839 | 1.259 | 1.154 | 0.839 | 0.839 |
| ReCoRD | 82.6 | 75.7 | 15.8 | 80.9 | 77.3 | 76.2 | 79.7 | 77.4 |
| Wikitext-2 | 15.3 | 18.85 | 1.35e5 | 15.69 | 21.9 | 24.09 | 17.56 | 18.18 |
| Average Acc | 42.36 | 40.49 | 28.97 | 42.26 | 39.38 | 39.24 | 41.48 | 39.90 |

Table I.3: The full results of W4/8A8 GPT-3$_{350M}$ using different data resources.

| Tasks | Random Data | Wikipedia | Original Training Data |
|---|---|---|---|
| HellaSwag | 33.9 | 35.5 | 35.3 |
| LAMBADA | 26.1 | 33.9 | 37.4 |
| TriviaQA | 0.088 | 0.972 | 0.23 |
| WebQs | 0.049 | 0.344 | 0.049 |
| Winogrande | 50.3 | 52.4 | 51.8 |
| PIQA | 59.3 | 62.4 | 61.8 |
| ARC (Challenge) | 22.6 | 23.3 | 23.6 |
| ARC (Easy) | 38.3 | 40.0 | 40.5 |
| ANLI R1 | 33.0 | 32.0 | 33.8 |
| ANLI R2 | 34.3 | 34.7 | 33.5 |
| ANLI R3 | 33.4 | 32.9 | 33.4 |
| OpenBookQA | 29.2 | 28.0 | 28.2 |
| RACE-h | 27.8 | 29.1 | 29.7 |
| BoolQ | 47.8 | 52.6 | 61.9 |
| Copa | 65.0 | 69.0 | 66.0 |
| RTE | 50.5 | 52.7 | 53.1 |
| WSC | 36.5 | 36.5 | 36.5 |
| MultiRC | 1.574 | 1.154 | 0.839 |
| ReCoRD | 64.9 | 69.5 | 68.5 |
| Wikitext-2 | 40.63 | 30.36 | 31.13 |
| Average Acc | 34.45 | 36.16 | 36.64 |

Table I.4: The full results of GPT-NeoX$_{20B}$.

| Tasks | W16A16 | W8A8/16 |
|---|---|---|
| HellaSwag | 71.4 | 71.2 |
| LAMBADA | 71.7 | 71.9 |
| TriviaQA | 25.8 | 25.9 |
| WebQs | 6.3 | 6.64 |
| Winogrande | 66.0 | 65.7 |
| PIQA | 77.7 | 78.3 |
| ARC (Challenge) | 41.0 | 42.2 |
| ARC (Easy) | 68.5 | 68.8 |
| ANLI R1 | 33.1 | 33.9 |
| ANLI R2 | 33.4 | 34.4 |
| ANLI R3 | 35.1 | 35.4 |
| OpenBookQA | 39.8 | 38.8 |
| RACE-h | 38.5 | 37.6 |
| BoolQ | 69.4 | 69.9 |
| Copa | 84.0 | 85.0 |
| RTE | 54.9 | 54.9 |
| WSC | 50.0 | 44.2 |
| MultiRC | 3.57 | 4.41 |
| ReCoRD | 88.3 | 88.0 |
| Average Acc | 50.45 | 50.38 |