# OpenReview forum: "ZeroQuant: Efficient and Affordable Post-Training Quantization for Large-Scale Transformers"
_NeurIPS.cc/2022/Conference — NeurIPS 2022 Accept_

### Official Review · Reviewer_zNp9 · 2022-07-10

**Rating:** 7
**Confidence:** 4
**Soundness:** 4 excellent
**Presentation:** 3 good
**Contribution:** 3 good

**Summary:**

This paper studies an important task to quantize very large transformer models leveraging three tricks, namely hardware friendly WA quantization scheme, layer-wise distillation, and an optimized backend. Results are fairly SOTA covering mainstream large nets on comprehensive datasets, with clear ablations.

**Questions:**

- For Fig. 2, what does Dequante mean?
- Eqn. 1 LKD, though mentioned but has other losses been tried?
- Table 2, the W4/8 A16 LKD improvement margins are observed smaller than other cases, any specific potential reasons?

Minor
- Though this is an NLP-oriented paper, how will the method scales potentially to ViTs?

**Limitations:**

These have been addressed well.

**Strengths And Weaknesses:**

Strength
+ This is a very solid paper in experimental fronts with all software-hardware perspectives highly optimized to push the frontier further.
+ The evaluation is very thorough.
+ The method scales well across datasets, and scenarios considering dataset access, etc.
+ The paper is very well written.

Weekness
- Though it works great, the deployed methods are known methods and tricks, with very limited new inspiring factors to the literature.

---

> ### Author Response · Authors · 2022-07-30
> **Response**
>
> Thanks a lot for taking the time to review our work and providing your constructive feedback. Our detailed response is listed below.
>
> * Q1: Though it works great, the deployed methods are known methods and tricks, with very limited new inspiring factors to the literature.
>
> A: Some techniques we used in the paper are known from the literature, e.g., group-wise quantization for weight from Q-BERT paper. However, other parts in the paper are still novel and they bring new opportunities, e.g., the layer-by-layer knowledge distillation and the system backend support. Besides, verifying the effectiveness of existing methods for ever-large models is also an important direction.
>
> * Q2: For Fig. 2, what does Dequante mean?
>
> A: Dequantization means mapping the INT32 number back to FP16/FP32 for the following operators, e.g., layer norm. Those operators are usually not compute-bounded, so using INT format won’t introduce latency reduction but can introduce significant accuracy loss.
>
> * Q3: Eqn. 1 LKD, though mentioned but has other losses been tried?
>
> A: We tried both L2 and L1 losses, and the results from those two losses are very similar. Below we provide the results of W4/8A8 for BERT-base. As can be seen, the accuracy of L1 loss and L2 loss based LKD is similar to each other. We will include this in the final revision.
>
> |                          |     CoLA     |     MNLI-m    |     MNLI-mm    |     MRPC           |     QNLI     |     QQP            |     RTE      |     SST-2    |     STS-B          |     Ave.     |
> |--------------------------|--------------|---------------|----------------|--------------------|--------------|--------------------|--------------|--------------|--------------------|--------------|
> |     FP16                 |     59.72    |     84.94     |     85.06      |     86.27/90.57    |     92.15    |     91.51/88.56    |     72.20    |     93.23    |     90.06/89.59    |     83.95    |
> |     W4/8A8, L2   loss    |     58.80    |     83.09     |     83.65      |     85.78/89.90    |     90.76    |     89.16/84.85    |     71.84    |     93.00    |     88.16/88.55    |     82.71    |
> |     W4/8A8, L1 loss      |     58.90     |     83.41     |     83.84      |     83.82/89.00    |     90.55    |     88.92/84.12    |     73.29    |     92.66    |     88.18/88.56    |     82.66    |
>
> * Q4: Table 2, the W4/8 A16 LKD improvement margins are observed smaller than other cases, any specific potential reasons?
>
> A: The main reason is that we use a fixed set of hyper-parameters for all experiments (which is mentioned in Line 243-244 of main text with detailed setting in Appendix B.2). If tuning is performed, the accuracy of LKD can be further improved. Please see Table E.1 in appendix for the tuned results.
>
> * Q5: Though this is an NLP-oriented paper, how will the method scales potentially to ViTs?
>
> A: Yes, our method can be applied to ViTs as well. We chose Huggingface’s ViT checkpoint finetuned on ImageNet-1K to test ZeroQuant (INT8) on ImageNet dataset. We provide both Top-1 and Top-5 accuracy in the table. The accuracy of ZeroQuant INT8 is comparable with the original FP32 baseline.
>
> Please also refer to the answer of Reviewer sGAq Q5 for more discussion about directly applying PTQ.
>
> |    ViT-base              |   Top-1 / Top-5      |
> |------------------------|-----------------------------|
> |     Baseline, FP32     |     81.430 /   96.010       |
> |     ZeroQuant, INT8    |     81.464 /   95.988       |

---

> > ### Comment · Reviewer_zNp9 · 2022-08-07
> > **Response**
> >
> > The response has addressed my concerns, and after reading rebuttal, modifications, and other reviews, I maintain my score on positive side.

---

### Official Review · Reviewer_e8j8 · 2022-07-10

**Rating:** 7
**Confidence:** 3
**Soundness:** 3 good
**Presentation:** 3 good
**Contribution:** 3 good

**Summary:**

This manuscript proposes a new post-training quantization framework for large Transformer-based language models (e.g., GPT-3). Specifically, ZeroQuant consists of hardware-constraint group-wise weight quantization and *kernel fusion* based token-wise activation quantization, low-cost layer-by-layer distillation (LGD), and optimized transformer kernels backend support. The proposed framework successfully quantizes the large language models into INT8 or INT4 representation with negligible performance degradation, which results in remarkable compression performance and real-time speedup.

**Questions:**

1. In Tab. 3, why are the results of Q-BERT and MP-PTQ [3] not compared? I think it's better to conduct a fair comparison between these two methods.
2. I wonder whether the authors are willing to release the backend scheduling codes since I think it will be of value to the community and industry.

[3] Bondarenko, Yelysei, Markus Nagel, and Tijmen Blankevoort. "Understanding and Overcoming the Challenges of Efficient Transformer Quantization." *Proceedings of the 2021 Conference on Empirical Methods in Natural Language Processing*. 2021.

**Limitations:**

See Weakness.

**Strengths And Weaknesses:**

### Strength

 1. This manuscript is well-motivated, targeting a relevant problem of modern large Transformer-based language model post-training quantization and deployment. The idea of combining group-wise weight quantization under hardware constraints and token-wise activation quantization with the optimized kernel is pretty reasonable. It has its practical value from the modern industrial aspect.
 2. The idea of LKD is simple but effective. The experiments also show its low-cost benefits, and the results of LKD w/o original training data are interesting to me.
 3. This manuscript looks solid with extensive experiments across many benchmarks and advanced large language models like BERT and GPT-3. The real-time speedups on A100 devices are reported.
 4. The overall presentation quality is good.

### Weakness

1. Group-wise weight quantization (GQ) was first proposed by Q-BERT [1], and ZeroQunt directly adopts it but with the hardware constraint, which weakens the technical novelty. As shown in Tab. 9, GQ is the core component that boosts performance. I wonder what the result of the TQ-only is, which is now missing in the ablation study? Do the benefits mostly come from GQ? Please complete the ablation study.
2. The proposed quantization-optimized Transformer kernels are built upon the CUTLASS library [2]. I suggest the authors include more illustrations of the scheduling framework and show some more in-depth study of the benefits brought by the proposed kernel optimization strategy. Besides, since the proposed deployment flow is targeted at GEMM, it will be a limitation for mobile devices which generally have no GPUs but CPUs.
3. I suggest the authors include some visualization results of the quantized models to support the claim that the proposed method can deal with the dynamic range of full-precision activations and weights.

[1] Shen, Sheng, et al. "Q-Bert: Hessian based ultra low precision quantization of bert." *Proceedings of the AAAI Conference on Artificial Intelligence*. 2020.

[2] NVIDIA blog. CUTLASS: Fast Linear Algebra in CUDA C++. https://developer.nvidia.com/blog/cutlass-linear-algebra-cuda/, December 2017.

---
#### Post Rebuttal:

I appreciate the detailed rebuttal from the authors. I have carefully read the rebuttal and other reviews, and I have no further concerns except for a conservative view about the technical novelty. As such, I decide to raise my rating to Accept and I believe this paper is a solid paper that will be of interest to the community and industry. I hope the authors could incorporate the valuable evaluations and the reviewers' suggestions into the final version.

---

> ### Author Response · Authors · 2022-07-30
> **Response Part #1**
>
> Thanks a lot for taking the time to review our work and providing your constructive feedback. Our detailed response is listed below.
>
> * Q1: Group-wise weight quantization (GQ) was first proposed by Q-BERT [1], and ZeroQunt directly adopts it but with the hardware constraint, which weakens the technical novelty. As shown in Tab. 9, GQ is the core component that boosts performance. I wonder what the result of the TQ-only is, which is now missing in the ablation study? Do the benefits mostly come from GQ? Please complete the ablation study.
>
> A: Thanks a lot for pointing this out. Besides directly adopting group quantization from Q-BERT, we also include the real GPU kernel implementation. Without the GPU kernel part, there is no real speedup benefit for group quantization. Meanwhile, without token quantization, the average accuracy of GQ is 66.52. TQ can further bring this number to 81.06 (14.54 accuracy gain, which is non-trivial). This already demonstrates the benefit of TQ.
>
> For W4/8 quantization, the main bottleneck is from weight quantization as explained in Line 138-143 for GPT-3-style model and Appendix C for BERT. As such, here we compare the results for W8A8 with different quantization schemes for weight and activation to demonstrate the effectiveness of GQ and TQ. From the table below, we can clearly see the benefit of token-wise quantization.
>
> |             |     GQ     |     TQ     |     CoLA     |     MNLI-m    |     MNLI-mm    |     MRPC           |     QNLI     |     QQP            |     RTE      |     SST-2    |     STS-B          |     Ave.     |
> |-------------|------------|------------|--------------|---------------|----------------|--------------------|--------------|--------------------|--------------|--------------|--------------------|--------------|
> |     FP16    |     --     |     --     |     59.72    |     84.94     |     85.06      |     86.27/90.57    |     92.15    |     91.51/88.56    |     72.20    |     93.23    |     90.06/89.59    |     83.95    |
> |     W8A8    |     No     |     No     |     56.06    |     79.99     |     81.06      |     75.49/79.67    |     87.35    |     89.92/86.82    |     48.38    |     92.66    |     86.58/86.44    |     77.41    |
> |     W8A8    |     Yes    |     No     |     59.84    |     80.25     |     81.37      |     83.82/89.18    |     91.23    |     91.32/88.14    |     70.04    |     93.12    |     88.66/88.80    |     82.31    |
> |     W8A8    |     No     |     Yes    |     60.77    |     84.65     |     84.92      |     85.29/89.86    |     91.84    |     91.52/88.56    |     71.84    |     93.46    |     89.89/89.50    |     83.87    |
>
> * Q2: The proposed quantization-optimized Transformer kernels are built upon the CUTLASS library [2]. I suggest the authors include more illustrations of the scheduling framework and show some more in-depth study of the benefits brought by the proposed kernel optimization strategy. Besides, since the proposed deployment flow is targeted at GEMM, it will be a limitation for mobile devices which generally have no GPUs but CPUs.
>
> A: Thanks for the great feedback. We are eager to bring more detail on this part, and we will include more in-depth information on how the inference system gets superior performance. We include a comparison of our optimized INT8 GEMM with CUBLAS’s INT8/FP16 GEMM in the answer of Review R9ah Q6. Please take a look and we will also include this in the final revision.
>
> The main focus of this paper was to speedup large scale models (from hundreds of millions of parameters to up to 20B parameters) on GPUs devices in cloud and data centers, therefore we did not have a system backend solution for mobile devices. As such, we agree our GeMM implementation cannot be used for small-scale systems where GPU is not available. However, the fusion strategy mentioned in the paper is still beneficial to those architectures to reduce the quantization overhead, and the quantization pipeline can also be useful for other workloads as well.

---

> > ### Author Response · Authors · 2022-07-30
> > **Response Part #2**
> >
> > * Q3: I suggest the authors include some visualization results of the quantized models to support the claim that the proposed method can deal with the dynamic range of full-precision activations and weights.
> >
> > A: Thanks a lot for the great suggestion. We followed your suggestion and made the analysis. Here we summarize the average of relative error (for both weight and activation) for BERT-base. Here we use weight’s formula as an illustration:
> > $$Average(\frac{abs(W_{quantize}-W_{ori})}{abs(W_{ori})+1e-8}).$$
> > The results are summarized in the table. The proposed methods in ZeroQuant can significantly reduce the relative error as compared to the standard quantization method. We will include the illustration of this analysis in the final revision.
> >
> > |                           |     Standard   Quantization    |     ZeroQuant’s group-wise   quantization or token-wise quantization    |
> > |---------------------------|--------------------------------|-------------------------------------------------------------------------|
> > |     Weight, INT8          |     0.10                       |     0.05                                                                |
> > |     Weight, INT4          |     0.73                       |     0.43                                                                |
> > |     Activation,   INT4    |     0.22                       |     0.13                                                                |
> >
> >
> > * Q4: In Tab. 3, why are the results of Q-BERT and MP-PTQ [3] not compared?
> >
> > A: Thanks a lot for the great suggestion. Since the original Q-BERT and MP-PTQ do not include BERT-large study, we implement our own code to produce the result of Q-BERT based on the paper’s description and use the open-sourced code https://github.com/qualcomm-ai-research/transformer-quantization to produce the result of MP-PTQ (we use --quant-dict "{'y': 16, 'h': 16, 'x': 16, 'P': 16, 'C': 16}"  and --quant-setup MSE_logits for STS-B task, and we use --quant-dict "{'y': 16, 'h': 16, 'x': 16}" for other tasks). The results are shown below. As can be seen, ZeroQuant achieves similar accuracy as Q-BERT but without any training cost.
> >
> > For MP-PTQ, we can reproduce the paper’s result for BERT-base models. However, for BERT-large, all tasks give random guess except CoLA (we tried three different kinds of checkpoints, including Huggingface’s checkpoint, our own checkpoints, and checkpoints trained using the MP-PTQ repo). As such, we did not include the rest of the results. After increasing the bit-precision for activation from 8 to 10 bits, the accuracy of MP-PTQ is significantly boosted. Therefore, the quantization error of activation is the main cause of MP-PTQ on BERT-large and this is primarily from the dynamic range of activations as what we discussed in Section 3 and Appendix C. Also, this result (i.e., MP-PTQ works for BERT-base but not BERT-large out of the box) is aligned with our reported PTQ results in Table 2 and 3, i.e., BERT-large’s W8A8 has lower accuracy than BERT-base’s W8A8, which means BERT-large is more sensitive to dynamic activation ranges.
> >
> > |                          |     CoLA     |     MNLI-m    |     MNLI-mm    |     MRPC           |     QNLI     |     QQP            |     RTE      |     SST-2    |     STS-B          |     Ave.     |     Ave. Time (s)    |
> > |--------------------------|--------------|---------------|----------------|--------------------|--------------|--------------------|--------------|--------------|--------------------|--------------|----------------------|
> > |     FP16                 |     63.35    |     86.65     |     85.91      |     87.99/91.62    |     92.24    |     91.08/88.08    |     74.01    |     93.46    |     90.34/90.11    |     85.03    |     --               |
> > |     Q-BERT, W8A8         |     62.86    |     86.26     |     86.43      |     86.27/90.38    |     92.07    |     91.59/88.66    |     73.73    |     93.92    |     89.87/89.54    |     84.83    |     7181             |
> > |     MP-PTQ, W8A8/16      |     61.48    |     --        |     --         |     --             |     --       |     --             |     --       |     --       |     --             |     --       |     --               |
> > | MP-PTQ, W8A10/16         |     62.6     |     85.68     |     85.68      |     87.25/91.22    |     91.37    |     90.51/87.09    |     72.56    |     93.00    |     85.2/87.07     |     83.90    |     10               |
> > |     ZeroQuant,   W8A8    |     63.38    |     86.52     |     85.64      |     87.75/91.50    |     92.31    |     91.09/88.05    |     72.56    |     93.35    |     90.45/90.19    |     84.81    |     0                |
> >
> > * Q5: Plan of releasing the backend scheduling codes since I think it will be of value to the community and industry.
> >
> > A: We are gradually cleaning the code and making the code open-source along the way. We will include the open-sourced code link in the final revision.

---

### Official Review · Reviewer_R9ah · 2022-07-12

**Rating:** 4
**Confidence:** 5
**Soundness:** 2 fair
**Presentation:** 2 fair
**Contribution:** 3 good

**Summary:**

The work uses zero-shot row and group-wise quantization that is optimized with fused kernels to perform inference on BERT and large transformers with more than 6B parameters. Layer-by-layer knowledge distillation and post-training quantization is applied to reduce the quantization model degradation.


**Questions:**

- how many seeds do you use for your BERT experiments? For reliable results at least 5 seeds are needed from my experience.
- I benchmarked raw Int8 vs FP16 and I cannot replicate the speedups that you get. What GEMMs are you using? The best comparison is with a recent cuBLAS version (gold standard) vs Int8 CUTLASS (which is very slow for me compared to cuBLAS, but it fast compared to FP16 CUTLASS GEMM)


**Limitations:**

The limitations are not addressed. While current GPUs only support Int8/Int4 Tensor cores, a discussion on FP8 transformers is missing.

**Strengths And Weaknesses:**

Strengths:
- a broad range of evaluations that go beyond BERT
- fused functions and open-source software. This takes a lot of effort and is a highly valuable contribution

Weaknesses:
- speedups benchmarks seem unrealistic. Raw GEMM comparisons within CUTLASS are often favorable when comparing Int8 vs FP16 for fused or infused operations, but comparison with cuBLAS usually yields small speedups.
- comparison to row/token quantization and row/token+column quantization are missing (Wu et al., 2020)[1]. This is the state-of-the-art for zero-shot quantization, it should be included for comparison with current methods.
- the paper does not explain the quantization with equations, it is difficult to follow which quantization scheme is actually used. It appears to be absolute maximum or symmetric quantization. But it could also be a form of truncated min-max quantization. A discussion on symmetric vs asymmetric quantization is missing. An evaluation with asymmetric quantization methods is missing (it might address the outliers found in the token statistics)
- It seems evaluation on BERT favored the author's method due to extensive hyperparameter search. The best result from the hyperparameter search was reported and no multiple seeds were run. This information is only found in the appendix but should be in the main paper. In my experience, at least 5 seeds are required for reliable results and it is best to use the same hyperparameters for both baseline methods and newly developed methods. Otherwise, results are often dominated by noise.

I am happy to increase my score by up to 2 if these weaknesses are addressed and other reviewers do not find other major issues with this work.

[1] Integer quantization for deep learning inference: Principles and empirical evaluation.

---

> ### Author Response · Authors · 2022-07-30
> **Response Part #1**
>
> Thanks a lot for taking the time to review our work and providing your constructive feedback. Our detailed response is listed below.
>
> * Q1: speedups benchmarks seem unrealistic. Raw GEMM comparisons within CUTLASS are often favorable when comparing Int8 vs FP16 for fused or infused operations, but comparison with cuBLAS usually yields small speedups.
>
> A: The speedup here is based on the end-to-end inference runtime for the different models and we are comparing the baseline (PyTorch FP16) with our fully optimized INT8 inference system. Note that PyTorch is using cuBLAS GeMM (verified through Nsight Systems profiling). Our improvement here is not only coming from the GeMM itself which uses INT8 tensor cores, but also the quantization/dequantization fusion strategy mentioned in the paper and other fusion methods, which fuse element-wise operations, matrix multiplications, transpositions, and reductions all into a single kernel. These can significantly reduce the number of kernel invocations as well as main memory access to reduce the main memory access latency. We will add the description of other fusion operators in the final revision.
>
> Please also see the answer of Q6 for the GeMM clarification.
>
> * Q2: comparison to row/token quantization and row/token+column quantization are missing (Wu et al., 2020)[1]. This is the state-of-the-art for zero-shot quantization, it should be included for comparison with current methods.
>
> A: Thanks a lot for pointing out this paper. First of all, we want to clarify a few things: (1) per-column weight quantization can be viewed as a special case for group quantization (one column per group). And this scheme may not result in GPU latency reduction since GPU uses block-wise computation; (2) Wu et al. mainly focus on the algorithm accuracy evaluation, and they did not provide real speedup results. However, our paper focuses on both algorithm and hardware constraint/implementation, and we provide real speedup solutions.
>
> Here, we include the comparison of ZeroQunat and the work of Wu el al. (use per-column for weight and per-row for tokens). The accuracy of ZeroQuant and the method from Wu et al. is similar to each other.
>
> |                          |         CoLA        |      MNLI-m    |     MNLI-mm    |         MRPC       |       QNLI     |         QQP        |       RTE      |      SST-2     |        STS-B       |      Ave.    |
> |--------------------------|:-------------------:|:--------------:|:--------------:|:------------------:|:--------------:|:------------------:|:--------------:|:--------------:|:------------------:|:------------:|
> |     FP16                 |         59.72       |      84.94     |      85.06     |     86.27/90.57    |      92.15     |     91.51/88.56    |      72.20     |      93.23     |     90.06/89.59    |     83.95    |
> |     Wu et al.,   INT8    |     59.62           |     84.86      |     85.09      |     85.78/90.20    |     91.93      |     91.48/88.48    |     71.12      |     93.23      |     90.02/89.53    |     83.73    |
> |     ZeroQuant,   INT8    |         59.59       |      84.83     |      85.13     |     86.03/90.39    |      91.98     |     91.45/88.46    |      71.12     |      93.12     |     90.09/89.62    |     83.75    |

---

> > ### Author Response · Authors · 2022-07-30
> > **Response Part #2**
> >
> > * Q3: the paper does not explain the quantization with equations, it is difficult to follow which quantization scheme is actually used. It appears to be absolute maximum or symmetric quantization. But it could also be a form of truncated min-max quantization. A discussion on symmetric vs asymmetric quantization is missing. An evaluation with asymmetric quantization methods is missing (it might address the outliers found in the token statistics)
> >
> > A: Sorry for the confusion. We put some of these details in Appendix A.2 due to the space limit. Particularly, we use uniform, absolute maximum, and symmetric quantization. We will move these details into the main context and add more explanations in the final revision.
> >
> > Asymmetric quantization introduces the zero-point (or bias term), which will affect the inference speed performance (this bias-term introduces the extra matrix-vector production). That’s the reason why in ZeroQuant we choose symmetric quantization as the quantization scheme. We did not directly provide asymmetric quantization but MP-PTQ [7] in Table 2 applied asymmetric activation quantization. To further demonstrate that asymmetric quantization cannot fully resolve the outlier issue (and with higher cost due to the extra matrix-vector production), we provided asymmetric quantization results for PTQ (W8A8) on BERT-base and Bert-large here.
> >
> > As can be seen, asymmetric quantization boosts the PTQ accuracy a lot but it is still >1 point lower than FP16 BERT-base. However, our ZeroQuant can close this gap to 0.25 without involving any extra matrix-vector production for the bias/zero-point term from asymmetric quantization. For BERT-large, asymmetric PTQ has ~2 points accuracy degradation and ZeroQuant only has 0.22.
> >
> > We will also include this in the final revision.
> >
> > |     BERT-base                |     CoLA     |     MNLI-m    |     MNLI-mm    |     MRPC           |     QNLI     |     QQP            |     RTE      |     SST-2    |     STS-B          |     Ave.     |
> > |------------------------------|--------------|---------------|----------------|--------------------|--------------|--------------------|--------------|--------------|--------------------|--------------|
> > |     FP16                     |     59.72    |     84.94     |     85.06      |     86.27/90.57    |     92.15    |     91.51/88.56    |     72.20    |     93.23    |     90.06/89.59    |     83.95    |
> > |     PTQ, INT8, Symmetric     |     56.06    |     79.99     |     81.06      |     75.49/79.67    |     87.35    |     89.92/86.82    |     48.38    |     91.40    |     86.58/86.44    |     77.41    |
> > |     PTQ, INT8, Asymmetric    |     59.94    |     80.58     |     81.54      |     84.80/89.67    |     92.00    |     91.42/88.30    |     71.12    |     92.89    |     89.34/89.30    |     82.72    |
> > |     ZeroQuant,   INT8        |     59.59    |     84.83     |     85.13      |     86.03/90.39    |     91.98    |     91.45/88.46    |     71.12    |     93.12    |     90.09/89.62    |     83.75    |
> > |     BERT-large               |              |               |                |                    |              |                    |              |              |                    |              |
> > |     FP16                     |     63.35    |     86.65     |     85.91      |     87.99/91.62    |     92.24    |     91.08/88.08    |     74.01    |     93.46    |     90.34/90.11    |     85.03    |
> > |     PTQ, INT8, Symmetric     |     60.57    |     75.69     |     76.94      |     81.13/84.93    |     88.49    |     84.04/74.35    |     46.93    |     91.74    |     62.57/55.77    |     73.54    |
> > |     PTQ, INT8, Asymmetric    |     60.52    |     86.16     |     85.84      |     85.29/89.40    |     92.11    |     90.09/86.16    |     64.98    |     92.89    |     89.40/89.38    |     83.04    |
> > |     ZeroQuant,   INT8        |     63.38    |     86.52     |     85.64      |     87.75/91.50    |     92.31    |     91.09/88.05    |     72.56    |     93.35    |     90.45/90.19    |     84.81    |

---

> > > ### Author Response · Authors · 2022-07-30
> > > **Response Part #3**
> > >
> > > * Q4: It seems evaluation on BERT favored the author's method due to extensive hyperparameter search. The best result from the hyperparameter search was reported and no multiple seeds were run. This information is only found in the appendix but should be in the main paper. In my experience, at least 5 seeds are required for reliable results and it is best to use the same hyperparameters for both baseline methods and newly developed methods. Otherwise, results are often dominated by noise.
> > >
> > > A: Thanks a lot for pointing this out. First, we want to clarify that all the results we reported in the main text do not use any hyper-parameter tuning. For W8A8, since there is no training involved, there is no tuning at all. Also, for W4/8A16 or W4/8A8, the main text results use a fixed set of hyperparameters, which means there is no hyper-parameter tuning. This part is mentioned in Line 243-244 of the main text with the detailed setting in Appendix B. The tuning results in Appendix E are used to demonstrate the low-cost of layer-by-layer knowledge distillation as well as for better comparison if someone applied hyperparameter search.
> > >
> > > For now, the results we used in the paper are based on a single seed. The main reason is that layer-by-layer knowledge distillation is a lightweight training procedure. As such, the weight won’t be significantly changed during LKD finetuning. To further alleviate the concern about variance/noise, we run the results for BERT-base with W4/8A16 and W4/8A8 with 5 seeds and report the results here. We follow the same setting as we used in the main text, i.e., Line 243-244. Particularly, for LKD, we use 100 iterations with batch size 32 and sequence length 128, and we fix the learning rate as 5e-6. For group-wise quantization, we choose 48 groups for all weight matrices.  We use seed {40, 41, 42, 43, 44} for this experiment. The results are shown below. As can be seen, the 5-seed average result is similar to what we have reported in the main text. We will include this and perform the five-seed experiment for other results in the final revision.
> > >
> > > |                                                            |     CoLA     |     MNLI-m    |     MNLI-mm    |     MRPC           |     QNLI     |     QQP            |     RTE      |     SST-2    |     STS-B          |     Ave.     |
> > > |------------------------------------------------------------|--------------|---------------|----------------|--------------------|--------------|--------------------|--------------|--------------|--------------------|--------------|
> > > |     FP16                                                   |     59.72    |     84.94     |     85.06      |     86.27/90.57    |     92.15    |     91.51/88.56    |     72.20    |     93.23    |     90.06/89.59    |     83.95    |
> > > |     ZeroQuant, W4/8A16,   single seed used in main text    |     58.50    |     83.16     |     83.69      |     84.80/89.31    |     90.83    |     88.94/84.12    |     70.04    |     92.78    |     88.49/88.67    |     82.35    |
> > > |     ZeroQuant, W4/8A16,   5 seeds                          |     58.64    |     83.15     |     83.74      |     84.65/89.22    |     90.87    |     89.08/84.39    |     70.40    |     92.82    |     88.48/88.65    |     82.43    |
> > > |     ZeroQuant, W4/8A8,   single seed used in main text     |     58.80    |     83.09     |     83.65      |     85.78/89.90    |     90.76    |     89.16/84.85    |     71.84    |     93.00    |     88.16/88.55    |     82.71    |
> > > |     ZeroQuant, W4/8A8,   5 seeds                           |     58.76    |     83.03     |     83.63      |     84.75/89.26    |     90.78    |     89.45/85.97    |     71.12    |     92.78    |     88.16/88.56    |     82.52    |
> > >
> > > * Q5: how many seeds do you use for your BERT experiments? For reliable results at least 5 seeds are needed from my experience.
> > >
> > > A: Please see answer of Q4 above.

---

> > > > ### Author Response · Authors · 2022-07-30
> > > > **Response Part #4**
> > > >
> > > > * Q6: I benchmarked raw Int8 vs FP16 and I cannot replicate the speedups that you get. What GEMMs are you using? The best comparison is with a recent cuBLAS version (gold standard) vs Int8 CUTLASS.
> > > >
> > > > A: Sorry for the confusion. First, the speedups we get are based on the end-to-end inference. Please see Q1 for more details. For the usage of INT8 CUTLASS, this piece of information is in Appendix D. We provide more details here and we will add it to the main text in the final revision.
> > > >
> > > > For the INT8 kernels, we run a profiling on different tiling schedules for thread-blocks, WARPs, and tensor-cores (WMMA instructions) in order to get the best latency based on the model dimension and batch size. Note that there are some paddings needed to keep the tenor shapes 16, 32, 64, or 128-aligned in order to maximize the performance when configuring the CUTLASS GeMM kernels. In our inference system, we choose the best scheduling for each application that minimizes the padding and gives the best performance. In our experiments, we see much better performance on the INT8 kernels for CUTLASS vs cuBLAS. Furthermore, we can fuse the dequantization step by modifying the CUTLASS kernels whereas this is not an option for cuBLAS library as it is not open-sourced.
> > > >
> > > > We included detailed profiling tables below for BERT-scale matrix sizes and GPT-J/GPT-NeoX-scale matrix sizes. The hardware used is the 40GB-A100. Here we use bsz to represent batch size, and seq to represent sequence length. The time is computed based on 1000 runs average time and the TFLOPS here means the Tera-Flops/s the hardware achieved. As we can see, our highly optimized CUTLASS INT8 GEMM is much faster than FP16/INT8 CUBLAS GEMM.
> > > >
> > > > |      BERT    |            |                     |     FP16 CUBLAS    |                 |     INT8 CUBLAS    |                 |     INT8 CUTLASS    |                 |
> > > > |--------------|------------|---------------------|--------------------|-----------------|--------------------|-----------------|---------------------|-----------------|
> > > > |     bsz      |     seq    |     Matrix   dim    |     time   (ms)    |     TFLOPS      |     time   (ms)    |     TFLOPS      |     time   (ms)     |     TFLOPS      |
> > > > |     1        |     128    |     768             |     0.02355        |     6.411796    |     0.021473       |     7.031874    |     0.009805        |     15.39947    |
> > > > |     1        |     128    |     1024            |     0.019223       |     13.9643     |     0.022616       |     11.86932    |     0.010406        |     25.79559    |
> > > > |     1        |     256    |     768             |     0.018297       |     16.50514    |     0.025432       |     11.87458    |     0.009844        |     30.67661    |
> > > > |     1        |     256    |     1024            |     0.019206       |     27.95322    |     0.060184       |     8.920457    |     0.010737        |     50.00222    |
> > > > |     64       |     128    |     768             |     0.049335       |     195.8768    |     0.153681       |     62.88129    |     0.035627        |     271.2449    |
> > > > |     64       |     128    |     1024            |     0.077738       |     220.9962    |     0.255213       |     67.31586    |     0.050085        |     343.0171    |
> > > > |     64       |     256    |     768             |     0.096026       |     201.2717    |     0.303941       |     63.58921    |     0.064837        |     298.0926    |
> > > > |     64       |     256    |     1024            |     0.15432        |     222.6525    |     0.500961       |     68.58771    |     0.092166        |     372.8039    |
> > > > |     GPT      |            |                     |                    |                 |                    |                 |                     |                 |
> > > > |     bsz      |     seq    |     Matrix   dim    |     time   (ms)    |     TFLOPS      |     time   (ms)    |     TFLOPS      |     time   (ms)     |     TFLOPS      |
> > > > |     1        |     1      |     4096            |     0.027091       |     1.23857     |     0.031635       |     1.060677    |     0.017066        |     1.966157    |
> > > > |     1        |     1      |     6144            |     0.066214       |     1.140199    |     0.073625       |     1.025434    |     0.036825        |     2.050146    |
> > > > |     1        |     8      |     4096            |     0.025938       |     10.34901    |     0.061419       |     4.37056     |     0.017056        |     15.73871    |
> > > > |     1        |     8      |     6144            |     0.064703       |     9.334616    |     0.175115       |     3.44904     |     0.036513        |     16.54146    |
> > > > |     1        |     16     |     4096            |     0.026499       |     20.25984    |     0.061636       |     8.710351    |     0.017051        |     31.48534    |
> > > > |     1        |     16     |     6144            |     0.065374       |     18.47757    |     0.175221       |     6.893921    |     0.036419        |     33.16868    |

---

### Official Review · Reviewer_sGAq · 2022-07-19

**Rating:** 6
**Confidence:** 4
**Soundness:** 3 good
**Presentation:** 3 good
**Contribution:** 3 good

**Summary:**

This paper provides a zero-shot and training-free quantization scheme for large-scale language-based Transformer models.

Specifically, they investigate the challenges of NLP Transformers and propose group-wise or token-wise quantization for weights and activations, respectively.

Also, the consider an affordable layer-by-layer knowledge distillation algorithm (LKD) without having to access the original training data, which is often not available.

In addition, they incorporate above contribution to build a highly optimized quantization system, which can perform real-device speedups.

**Questions:**

- How about vision transformers (ViTs), will these family of models also suffer form the same problem and therefore can use the proposed techniques?

- The distillation may require even larger models? Which one are you considering? What if we use smaller models to distill the larger models when applying quantization?

- Any opensource idea for the quantization itself? A miminalist codebase that provide soly quantitzation function will make it more convincing.

- A literature review of current staus of quantization on large-scale language models are expected, e.g., a roadmap of the past few years and which point does this paper fit in.

I am open to further boost the score given good answers/rebuttals.

**Limitations:**

No negative societal impact.

**Strengths And Weaknesses:**

Strengths

- I am glad to see this post training quantization method can achieve real device speedups, i.e., not fake quantization with dequantization involved for mimicing the behaviour on general GPU devices.
- The authors are working on large models, which often require a lot of GPU resources to train or retrain. So post training quantization (and data free quantization) is expected. Such a motivation makes a lot of sense to me.

Weaknesses

- The author put a lot of benchmark results while I am somehow get lost how many accuracy drops (a range there will be better) are expected when applying your proposed quantization techniques?
- Will the proposed techniques also be generalized to small Transformer models? How about CNN models, will there are any differences among them when applying the quantization scheme?
- How do you handle the softmax functions in the Transformer models? I assume that it will need more specific step as its quantization will lead to more accuracy degradation as I herad form recent publications.

---

> ### Author Response · Authors · 2022-07-29
> **Response Part #1**
>
> Thanks a lot for taking the time to review our work and providing your constructive feedback. Our detailed response is listed below.
>
> * Q1: The author put a lot of benchmark results while I am somehow get lost how many accuracy drops (a range there will be better) are expected when applying your proposed quantization techniques?
>
> A: Thank you so much for pointing out the confusion. The main reason we did not provide an overall accuracy degradation range is as you pointed out: we have many different benchmarks and different scenarios (like encoder model with finetuning and decoder model with zero-shot evaluation, as well as INT8 and mixed INT4/INT8 quantization). The accuracy drop varies a lot under different benchmarks and scenarios. Here, we give a summary according to those settings:
>
> (1)	For BERT on GLUE benchmark, the accuracy degradation is usually within 0—0.2 for W8A8 and about 1--2 (except RTE for BERT-large) for W4/8A8 (Table 2, 3, E1, and E2 in the paper).
>
> (2)	For GPT-3-style models (350M and 1.3B) with W8A8, the average accuracy degradation is about 0.1% and the perplexity drop is about 0.3 points. However, for each of individual tasks, due to the nature of zero-shot evaluation (no task-specific fine-tuning, different number of eval cases between tasks, etc), the accuracy change varies. Particularly, for some cases, W8A8 even outperforms FP16.
>
> (3)	With W4/8A8, (a) for the 350M GPT-3-style model, the average accuracy degradation is about 2.3% and the perplexity drop is about 9.5; (b) for the 1.3B model, the average accuracy degradation is 2.5% and the perplexity drop is about 2.5. Note that those results are achieved without tuning the layer-by-layer knowledge distillation.
>
> (4)	For GPT-J (6B), our INT8 model has about 0.3 average perplexity loss; and for GPT-NeoX (20B), our INT8 model has about 0.08% accuracy loss.
>
> We will include this summary in the revised version.
>
> * Q2: Will the proposed techniques also be generalized to small Transformer models?
>
> A: Yes, our proposed method can also be applied to small Transformer models as well. One example is BERT-base used in the paper. To further demonstrate this, we use TinyBERT (4L-312H version using Huggingface model zoo’s checkpoint from user Sayan01) with ZeroQuant W8A8 (without LKD) as an example (The checkpoint of CoLA is broken so we did not include the result of CoLA):
>
> |                           |     CoLA     |     MNLI-m    |     MNLI-mm    |        MRPC      |     QNLI    |           QQP          |      RTE    |     SST-2    |       STS-B      |
> |:-------------------------:|:------------:|:-------------:|:--------------:|:----------------:|:-----------:|:----------------------:|:-----------:|:------------:|:----------------:|
> |            FP16           |       --     |      80.1     |       80.2     |     83.3/88.8    |     83.7    |        88.5/85.1       |     66.8    |      87.0    |     87.3/87.1    |
> |     ZeroQuant   (W8A8)    |       --     |      80.1     |       79.9     |     82.1/88.2    |     84.1    |     88.7/85.2          |     67.5    |      86.9    |     87.6/87.4    |
>
> As can be seen, the performance of ZeroQuant (INT8) is comparable with the original FP16 baseline. We will include this example in the revised version.

---

> > ### Author Response · Authors · 2022-07-30
> > **Response Part #2**
> >
> > * Q3: How about CNN models, will there are any differences among them when applying the quantization scheme?
> >
> > A: We can still apply our quantization scheme on CNN models with some modification. For instance, we can use per-output-channel quantization for convolutional kernels as well as per-image quantization for hidden states (images). As such, this quantization is still satisfied with the hardware compute constraint since we do not break the computation granularity of GPUs (i.e., real speedup can still be achieved on GPUs). To verify this, we applied this scheme for ResNet18/ResNet50/ResNet152 with Top-1 and Top-5 accuracy. The code base is from https://github.com/pytorch/examples/blob/main/imagenet/main.py. As can be seen, the accuracy degradation of ZeroQuant is ~0.1% as compared to FP32 models.
> >
> >
> > |                        |     ResNet18             |     ResNet50           |     ResNet152            |
> > |:----------------------:|--------------------------|------------------------|--------------------------|
> > |     Baseline, FP32     |     69.766 /   89.068    |     76.132 / 92.862    |     78.318 / 94.050      |
> > |     ZeroQuant, INT8    |     69.686 /   89.004    |     76.022 / 92.848    |     78.272 /   94.002    |
> >
> >
> >
> > * Q4: How do you handle the softmax functions in the Transformer models? I assume that it will need more specific steps as its quantization will lead to more accuracy degradation as I herad form recent publications.
> >
> > A: That’s a great point. We keep Softmax and GeLU in FP16. Quantizing those non-GeMM operators will not introduce much speedup since they are not the bottleneck for computation and they are easy to be fused with other operators (e.g., like bias-gelu operator). However, as you pointed out, quantizing them will introduce more accuracy degradation. As such, we do not quantize them in ZeroQuant.
> >
> > * Q5: How about vision transformers (ViTs), will these family of models also suffer form the same problem and therefore can use the proposed techniques?
> >
> > A: Yes, our method can be applied to ViTs as well. We chose Huggingface’s ViT checkpoint finetuned on ImageNet-1K to test ZeroQuant (INT8) and PTQ (as described in the paper). We provide both Top-1 and Top-5 accuracy in the table.
> > We analyzed the range of weight and the range of activations and found out that the dynamic range issue of ViT is better than those language-based models (BERT, GPT-style models). As such the accuracy degradation of ViT is better than that of BERT/GPT-style models. Particularly, the accuracy of PTQ has 1.32% Top-1 accuracy drop as compared to baseline. However, ZeroQuant INT8 can achieve comparable accuracy as the original FP32 baseline (0.034% better for Top-1 and 0.022 worse for Top-5). In summary, ZeoQuant is still beneficial for ViT.
> >
> > We will include this result in the final revision.
> >
> > |    ViT-base        |       Top-1 / Top-5         |
> > |------------------------|--------------------------|
> > |     Baseline, FP32     |     81.430 /   96.010    |
> > |     PTQ, INT8          |     80.110 /   95.318    |
> > |     ZeroQuant, INT8    |     81.464 /   95.988    |

---

> > > ### Author Response · Authors · 2022-07-30
> > > **Response Part #3**
> > >
> > > * Q6: The distillation may require even larger models? Which one are you considering? What if we use smaller models to distill the larger models when applying quantization?
> > >
> > > A: Sorry for the confusion. We did not introduce extra teacher models to do distillation but used the unquantized model as the teacher. For example, in order to quantize BERT-base, we have the original unquantized model (BERT-base-FP16). As such, we can use this BERT-base-FP16 as the teacher to do the layer-by-layer knowledge distillation (LKD). Since we quantize the network one layer at a time, we only need extra memory for the weight, activation, and optimizer states for one specific layer for LKD. This can significantly save the compute cost and memory cost. We will clarify this in the final revision.
> > >
> > > * Q7: Any opensource idea for the quantization itself? A miminalist codebase that provide soly quantitzation function will make it more convincing.
> > >
> > > A: We are gradually cleaning the code and making the code open-source along the way. We will include the open-sourced code link in the final revision.
> > >
> > > * Q8: A literature review of current staus of quantization on large-scale language models are expected, e.g., a roadmap of the past few years and which point does this paper fit in
> > >
> > > A: We will add a discussion in the related work section, talking about how the current work fits to the landscape of quantization for large-scale language models from the following three aspects:
> > >
> > > (1) Existing works such as [3, 22, 57, 77, 80, …] (use the same reference indexes as those in the paper) often focuses on encoder models such as BERT, while we study both encoder and decoder models with emphasis on decoders (e.g., GPT style models), which are currently the largest models and most expensive models to compress.
> > >
> > > (2) Existing works often focus on their evaluation on a limited model scale, e.g., BERT-base/large have ~110M/350M parameters, which are >50 times smaller than the largest model we compress in this work (20B GPT-NeoX). As such, various expensive optimization techniques such as retraining/fine-tuning and/or knowledge distillation are used to recover the accuracy loss in those work [3, 30, 62, 80]. However, compression cost becomes a major bottleneck for compressing giant multi-billion scale models and requires additional considerations on the compression efficiency and cost. Our ZeroQuant provides an effective and lightweight solution to this raising challenge.
> > >
> > > (3) Existing works on quantizing large-scale language models primarily focus on studying the quantization algorithm impact to the model accuracy and do not discuss how to deal with the potential high cost associated with quantization/dequantization. In contrast, we provide the corresponding system optimizations such as fusions to obtain end-to-end latency speedups.

---

### Meta-Review · Area_Chair_4iyz · 2022-08-26

**Recommendation:** Accept
**Confidence:** Certain

**Metareview:**

This paper explores post-training quantization on large transformer-based models, using techniques such as group-wise weight quantization, token-wise activation quantization and layer-by-layer distillation. The paper also provides a system backend support that demonstrate speedup on commercial GPU devices.

Overall, this is a solid paper. The quantization methods used in the paper are not exactly novel. Not only the group-wise and token-wise quantization as pointed out by the reviewers, but also the layer-by-layer distillation using high precision models have been reported recently (e.g. BRECQ [ICLR2021] and AdaQuant[ICML2021]). However, it is appreciated that the authors could take these techniques further to 1) implement and optimize GPU kernels to demonstrate real speedup and 2) evaluate the techniques on large scale models. With the open-source code (as promised), I think the research community and industry can all benefit from this work.

In addition, the paper is well written and easy to follow. The methods are evaluated thoroughly. During the rebuttal period, the authors provide very detailed response to address the questions and concerns raised by the reviewers. Therefore, this paper is recommended for acceptance.


**Award:**

No

---

### Decision · Program_Chairs · 2022-09-14

Accept